# Comparing and Contrasting Deep Learning Weather Prediction Backbones on Navier-Stokes and Atmospheric Dynamics

## Abstract

Remarkable progress in the development of Deep Learning Weather Prediction (DLWP) models positions them to become competitive with traditional numerical weather prediction (NWP) models. Indeed, a wide number of DLWP architectures—based on various backbones, including U-Net, Transformer, Graph Neural Network (GNN), and Fourier Neural Operator (FNO)—have demonstrated their potential at forecasting atmospheric states. However, due to differences in training protocols, forecast horizons, and data choices, it remains unclear which (if any) of these methods and architectures are most suitable for weather forecasting and for future model development. Here, we step back and provide a detailed empirical analysis, under controlled conditions, comparing and contrasting the most prominent DLWP models, along with their backbones. We accomplish this by predicting synthetic two-dimensional incompressible Navier-Stokes and real-world global weather dynamics. In terms of accuracy, memory consumption, and runtime, our results illustrate various tradeoffs. For example, on synthetic data, we observe favorable performance of FNO; and on the real-world WeatherBench dataset, our results demonstrate the suitability of ConvLSTM and SwinTransformer for short-to-mid-ranged forecasts. For long-ranged weather rollouts of up to 365 days, we observe superior stability and physical soundness in architectures that formulate a spherical data representation, i.e., GraphCast and Spherical FNO. In addition, we observe that all of these model backbones "saturate," i.e., none of them exhibit so-called neural scaling, which highlights an important direction for future work on these and related models. The code is available at https://github.com/amazon-science/dlwp-benchmark.

## 1 Introduction

Deep Learning Weather Prediction (DLWP) models have recently evolved to form a promising and competitive alternative to numerical weather prediction (NWP) models (Kalnay, 2003; Bauer et al., 2015; Dueben and Bauer, 2018). In early attempts, Scher and Messori (2018); Weyn et al. (2019) designed U-Net models (Ronneberger et al., 2015) on a cylinder mesh, learning to predict air pressure and temperature dynamics on a coarse global resolution of $5.625°$. More recently, Pathak et al. (2022) proposed FourCastNet on basis of the Adaptive Fourier Neural Operator (AFNO) (Guibas et al., 2021)—an efficient formulation of Li et al. (2020b)'s FNO—deploying the native $0.25°$ resolution of the ERA5 reanalysis dataset (Hersbach et al., 2020), which covers the globe with $721 \times 1440$ data points. The same dataset finds application in the Vision Transformer (ViT) (Dosovitskiy et al., 2020) based Pangu-Weather model (Bi et al., 2023) and the message-passing Graph Neural Network (GNN) (Battaglia et al., 2018; Pfaff et al., 2020; Fortunato et al., 2022) based GraphCast model (Lam et al., 2022).

In a comparison of state-of-the-art (SOTA) DLWP models, Rasp et al. (2023) find that GraphCast generates the most accurate weather forecasts on lead times up to ten days. GraphCast was trained on 221 variables from ERA5—substantially more than the 67 and 24 prognostic variables considered in Pangu-Weather and FourCastNet. The root of GraphCast's improved performance, though, remains entangled in details of the architecture type, choice of prognostic variables, and training protocol. Here, we seek to elucidate the effect of DLWP architectures' backbones, i.e., GNN, Transformer,

U-Net, or Fourier Neural Operator (FNO) (Li et al., 2020b). To this end, we first design a benchmark on two-dimensional Navier-Stokes simulations to train and evaluate various architectures, while controlling the number of parameters to generate cost-performance tradeoff curves. We then expand the study from synthetic to real-world weather data provided through WeatherBench (Rasp et al., 2020). WeatherBench was recently extended to WeatherBench2 (Rasp et al., 2023) and compares SOTA DLWP. An end-to-end comparison of DLWP architectures controlling for parameter count, training protocol, and set of prognostic variables, has not been performed. This lack of controlled experimentation hinders the quality assessment of backbones used in DLWP (and potentially beyond in other areas of scientific machine learning). Addressing this issue in a systematic manner is a main goal of our work.

With our analysis, we also seek to motivate architectures that have the greatest potential in addressing downsides of current DLWP models. To this end, we focus on three aspects: (1) short- to mid-ranged forecasts out to 14 days; (2) stability of long rollouts for climate lengthscales; and (3) physically meaningful predictions. Our aim is to help the community find and agree on a suitable DLWP backbone and to provide a rigorous benchmarking framework that facilitates a fair model comparison and supports architecture choices for dedicated forecasting tasks.

We find that FNO reproduces the Navier-Stokes dynamics most accurately, followed by SwinTransformer and ConvLSTM. In addition, we make the following observations on WeatherBench:

- Over short- to mid-ranged lead times—aspect (1) of WeatherBench—we observe a surprising forecast accuracy of ConvLSTM (the only recurrent and oldest architecture in our comparison), followed by SwinTransformer and FourCastNet.
- In terms of stability (2), explicit model designs tailored to weather forecasting are beneficial, e.g., Pangu-Weather, GraphCast, and Spherical FNO.
- Similarly, these same three sophisticated DLWP models reproduce characteristic wind patterns (3) more accurately than pure backbones (U-Net, ConvLSTM, SwinTransformer, FNO) by better satisfying kinetic energy principles.

While we identify no strict one-fits-all winner model, the strengths and weaknesses of the benchmarked architectures manifest in different tasks. Also, although targeting neural scaling behavior was not the main focus of this work, we observe that the performance improvement of all of these models saturates (as model, data, or compute are scaled). This highlights an important future direction for making model backbones such as these even more broadly applicable for weather prediction and beyond.

## 2 OUR APPROACH, RELATED WORK, AND METHODS

We compare five model classes that form the basis for SOTA DLWP models and include four established DLWP models in our analysis. In the following, we provide a brief overview of these nine methods. See Appendix A.1.1 for more details, and see Table 2 in that appendix for how we modify these methods to vary the number of parameters. As a naïve baseline and upper bound for our error comparison, we implement `Persistence`,[1] which predicts the last observed value as a constant over the entire forecast lead time. For short lead times in the nowcasting range (out to 6 hours, depending on the variable), this baseline is considered a decent strategy in atmospheric science that is not trivial to beat (Murphy, 1992). On WeatherBench, we include `Climatology` forecasts, which represent the averaged monthly observations from 1981 to 2010.

Starting with early deep learning (DL) methods, we include convolutional long short-term memory (`ConvLSTM`) (Shi et al., 2015), which combines spatial and temporal information processing by replacing the scalar computations of LSTM gates (Hochreiter and Schmidhuber, 1997) with convolution operations. `ConvLSTM` is one of the first DL models for precipitation nowcasting and other spatiotemporal forecasting tasks, and it finds applications in Google's MetNet1 and MetNet2 (Sønderby et al., 2020; Espeholt et al., 2022). Among early DL methods, we also benchmark `U-Net`, which is one of the most prominent and versatile DL architectures. It was originally designed for biomedical image segmentation (Ronneberger et al., 2015), and it forms the backbone of many DLWP (and other) models (Weyn et al., 2019; 2020; 2021; Karlbauer et al., 2023; Lopez-Gomez et al., 2023).

---

[1] In the following, we denote models that are included in our benchmark with `teletype font`.

We include two more recent architecture backbones, which power SOTA DLWP models based on Transformers (Bi et al., 2023) and GNNs (Lam et al., 2022). The Transformer architecture (Vaswani et al., 2017) has found success with image processing (Dosovitskiy et al., 2020), and it has been applied to weather forecasting, by viewing the atmospheric state as a sequence of three-dimensional images (Gao et al., 2022). `Pangu-Weather` (Bi et al., 2023, by Huawei) and FuXi (Chen et al., 2023b) use the `SwinTransformer` backbone (Liu et al., 2021) and add a Latitude-Longitude representation. Similarly, FengWu (Chen et al., 2023a; Han et al., 2024) use Transformers, like Microsoft when designing ClimaX for weather and climate related downstream tasks (Nguyen et al., 2023). ClimaX introduces a weather-specific embedding to treat different input variables adequately, which also finds application in Stormer (Nguyen et al., 2024). Multi-Scale MeshGraphNet (`MS MeshGraphNet`) (Fortunato et al., 2022) extends Pfaff et al. (2020)'s MeshGraphNet—a message-passing GNN processing unstructured meshes—to operate on multiple grids with different resolutions. `MS MeshGraphNet` forms the basis of `GraphCast` (Lam et al., 2022) using a hierarchy of icosahedral meshes on the sphere.

Lastly, we benchmark architectures based on `FNO` (Li et al., 2020b). `FNO` is a type of operator learning method (Li et al., 2020a; Lu et al., 2021; Gupta et al., 2021) that learns a function-to-function mapping by combining pointwise operations in physical space and in the wavenumber/frequency domain. Along with `FNO`, Li et al. (2020b) propose a In contrast to the aforementioned architectures, `FNO` is a discretization invariant operator method. While `FNO` can be applied to higher resolutions than it was trained on, it may not be able to predict processes that unfold on smaller scales than observed during training (Krishnapriyan et al., 2023). These uncaptured small-scale processes can be important in turbulence modeling. We implement a two- and a three-dimensional variant of `FNO`, as specified in Appendix A.1.1. We also experiment with `TFNO`, which uses a Tucker-based tensor decomposition (Tucker, 1966; Kolda and Bader, 2009) to be more parameter efficient. `FNO` serves as the basis for LBNL's and NVIDIA's `FourCastNet` series (Pathak et al., 2022; Bonev et al., 2023; Kurth et al., 2023). In particular, we consider both the original `FourCastNet` implementation based on Guibas et al. (2021) and the newer Spherical Fourier Neural Operator (`SFNO`) (Bonev et al., 2023), which works with spherical data and is promising for weather prediction on the sphere.

## 3 EXPERIMENTS AND RESULTS

In the following Section 3.1, we start with controlled experimentation on synthetic Navier-Stokes data. In Section 3.2, we extend the analysis to real-world weather data from WeatherBench, featuring a subset of variables from the ERA5 dataset (Hersbach et al., 2020). ERA5 is the reanalysis product from the European Centre of Medium-Ranged Weather Forecasts (ECMWF), and it is a result of aggregating observation data into a homogeneous dataset using NWP models.

### 3.1 SYNTHETIC NAVIER-STOKES SIMULATION

We conduct three series of experiments to explore the ability of the architectures (see Section 2) to predict the two-dimensional incompressible Navier-Stokes dynamics in a periodic domain. We choose Navier-Stokes dynamics as they find applications in NWP[2] and can provide insights on how each model may perform on actual weather data.[3] Concretely, in the three experiments, we address the following three questions:

(1) Which DLWP backbone is most suitable for predicting less turbulent spatiotemporal Navier-Stokes dynamics with small Reynolds Numbers, according to the RMSE metric? (Section 3.1.1).

(2) Do the results of Experiment 1 (the model ranking when predicting Navier-Stokes dynamics) hold for larger Reynolds Numbers, i.e., on more turbulent data? (Section 3.1.2).

(3) How does the size of the dataset effect each model and the ranking of all models? (Section 3.1.3).

---

[2]When simulating density and particle propagation in the atmosphere, NWP models solve a system of equations in each grid cell under consideration of the Navier-Stokes equations, among others, to conserve momentum, mass, and energy (Bauer et al., 2015).

[3]A direct transfer of the results from Navier-Stokes to weather dynamics is limited, as our synthetic data only partially represents rotation or mean flow characteristics and does not encompass the multi-scale complexity present in true atmospheric flows.

Table 1: RMSE scores for experiment 1, reported for each model under different number of parameters. Errors reported in italic correspond to models that were trained with gradient clipping (by norm) due to stability issues. With `OOM` and `sat`, we denote models that ran out of GPU memory and saturated, respectively. Saturated means that we did not further increase the parameters because the performance already saturated over smaller parameter ranges. Best results are shown in bold.

| Model | #params | | | | | | | | |
|---|---|---|---|---|---|---|---|---|---|
| | 5 k | 50 k | 500 k | 1 M | 2 M | 4 M | 8 M | 16 M | 32 M |
| Persistence | .5993 | .5993 | .5993 | .5993 | .5993 | .5993 | .5993 | .5993 | .5993 |
| ConvLSTM | .1278 | .0319 | .0102 | *.0090* | *.2329* | *.4443* | OOM | — | — |
| U-Net | .5993 | .0269 | .0157 | .0145 | .0131 | .0126 | .0126 | sat | — |
| FNO3D L1-8 | .3650 | .2159 | .1125 | .1035 | .1050 | .0383 | .0144 | .0095 | — |
| TFNO3D L1-16 | — | — | — | .0873 | .0889 | .0221 | .0083 | .0066 | .0069 |
| TFNO3D L4 | — | .0998 | .0173 | .0127 | .0107 | .0091 | .0083 | sat | — |
| TFNO2D L4 | **.0632** | **.0139** | **.0055** | **.0046** | **.0043** | .0054 | **.0041** | **.0046** | sat |
| SwinTransformer | .1637 | .0603 | .0107 | .0084 | .0070 | OOM | — | — | — |
| FourCastNet | .1558 | .0404 | .0201 | .0154 | *.0164* | *.0153* | *.0149* | sat | — |
| MS MeshGraphNet | *.2559* | *.0976* | *.5209* | OOM | — | — | — | — | — |

We discretize our data on a two-dimensional $64 \times 64$ grid, and we design the experiments to test two levels of difficulties by generating less and more turbulent data, with Reynolds Numbers $Re = 1 \times 10^3$ (experiment 1) and $Re = 1 \times 10^4$ (experiments 2 and 3), respectively. For experiments 1 and 2, we generate $1$ k samples. Experiment 3 repeats experiment 2 with an increased number of $10$ k samples. Our experiments are designed to test: (1) easier vs. harder problems, with the modification in $Re$; and (2) the effect of the dataset size.

For comparability, the initial condition and forcing of the data generation process are chosen to be identical with those in Li et al. (2020b); Gupta et al. (2021) (see Appendix A.1.2). Also, following Li et al. (2020b), the models receive a context history of $h = 10$ input frames, on basis of which they autoregressively generate the remaining 40 (experiment 1) or 20 (experiments 2 and 3) frames.[4] Concretely, we apply a rolling window when generating autoregressive forecasts, by feeding the most recent $h$ frames as input and predicting the next single frame, i.e., $\hat{y}_{t+1} = \varphi_\theta(x_{t-h,...,t})$, where $\hat{y}_{t+1}$ denotes the prediction of the next frame generated by model $\varphi$ with trainable parameters $\theta$, and $x_{t-h,...,t}$ denotes the most recent $h$ frames provided as input concatenated along the channel dimension. The three-dimensional (T)FNO models make an exception to the autoregressive rolling window approach, by receiving the first $h$ frames $x_{0:h}$ as input to directly generate a prediction $\hat{y}_{h+1:T}$ of the entire remaining sequence in a single step. See Appendix A.1.3 for our training protocol featuring hyperparameters, learning rate scheduling, and number of weight updates.

### 3.1.1 EXPERIMENT 1: SMALL REYNOLDS NUMBER, 1 K SAMPLES

In this experiment, we generate less turbulent dynamics with Reynolds Number $Re = 1 \times 10^3$ and a sequence length of $T = 50$. The root mean squared error (RMSE) metric, reported in Table 1 and Figure 1 (left) shows that TFNO2D performs best, followed by TFNO3D, SwinTransformer, FNO3D, ConvLSTM, U-Net, FourCastNet, and MS MeshGraphNet (see qualitative results in Figure 6 in Appendix A.2.1 with the same findings). All models outperform the naïve Persistence baseline, which predicts the last observed state, i.e., $\hat{y}_t = x_h$. This principally indicates a successful training of all models. We observe substantial differences between models in the error saturation when increasing the number of parameters, which supports the ordering of architectures seen in Figure 6. Concretely, with an error of $1 \times 10^{-2}$, MS MeshGraphNet does not reach the accuracy level of other models. Beyond $500$ k parameters, the model hits the memory constraint and also does not converge.[5] We identify remarkable effects of the graph design by comparing periodic 4-stencil, 8-stencil, and Delaunay triangulation graphs. The latter supports a stable convergence most (see Figure 7 in Appendix A.2.1 for details). Throughout our experiments, we use the 4-stencil graph.

---

[4]Larger Reynolds Numbers lead to more turbulent dynamics that are harder to predict. Thus, Li et al. (2020b) select $T = 50$ and $T = 30$ for $Re = 1e3$ and $Re = 1e4$, respectively. We follow this convention.

[5]Experiments are performed on two AWS `g5.12xlarge` instances, featuring four NVIDIA A10G GPUs with 23 GB RAM each. We use single GPU training throughout our experiments.

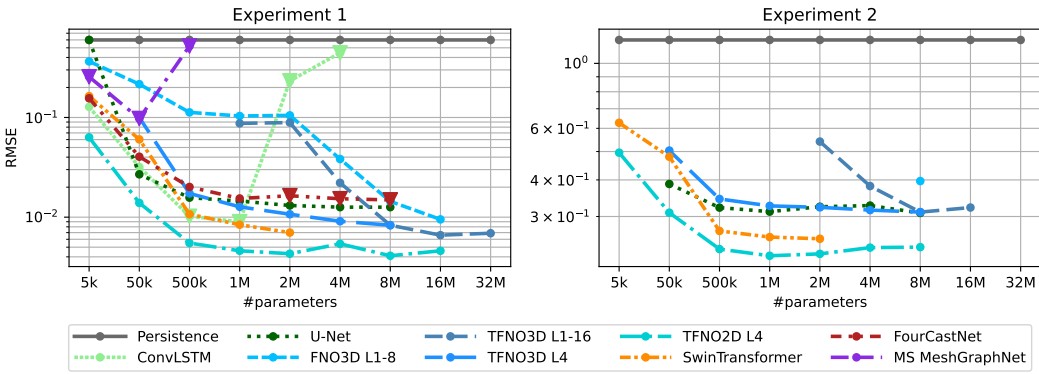

Figure 1: RMSE vs. number of parameters for models trained on Reynolds Numbers $Re = 1 \times 10^3$ (experiment 1, left) and $Re = 1 \times 10^4$ (experiment 2, right) with 1k samples. Note the different y-axis scales. Triangle markers indicate models with instability issues during training, requiring the application of gradient clipping. In the limit of growing parameters, each model converges to an individual error score (left), which seems consistent across data complexities (cf. left and right).

`ConvLSTM` is competitive within the low-parameter regime, saturating around an RMSE of $9 \times 10^{-3}$; yet, the model becomes unstable with large channel sizes (which we could not compensate even with gradient clipping). It runs out of memory beyond $4\,\mathrm{M}$ parameters, and suffers from exponential runtime complexity (see Figure 8, right, in Appendix A.2.1). `SwinTransformer` generates comparably accurate predictions, reaching an error of $7 \times 10^{-3}$, before quickly running out of memory when going beyond $2\,\mathrm{M}$ parameters. `U-Net` and `FourCastNet` exhibit a similar behavior, saturating at the $1\,\mathrm{M}$ parameter configuration and reaching error levels of $1.2 \times 10^{-2}$ and $1.5 \times 10^{-2}$, respectively. In `FNO3D` and the Tucker tensor decomposed `TFNO3D` (Kolda and Bader, 2009), we observe a two-staged saturation, where the models first converge to a poor error regime of $1 \times 10^{-1}$, albeit approaching a remarkably smaller RMSE of $9 \times 10^{-3}$ and $6 \times 10^{-3}$, respectively, when increasing the number of *layers* from 1 at `#params` $\leq 2\,\mathrm{M}$ to 2, 4, 8, and 16 to obtain the respective larger parameter counts.[6] Instead, when fixing the numbers of layers at $l = 4$ and varying the number of channels in `TFNO3D L4`, we observe better performance compared to the single-layer `TFNO3D L1-16` in the low-parameter regime (until $2\,\mathrm{M}$ parameters), albeit not competitive with other models. To additionally explore the effect of the number of layers vs. channels in `TFNO3D`, we vary the number of parameters either by increasing the layers over $l \in [1, 2, 4, 8, 16]$, while fixing the number of channels at $c = 32$ in `TFNO3D L1-16`, or by increasing the number of channels over $c \in [2, 8, 11, 16, 22, 32]$ while fixing the number of layers at $l = 4$ in `TFNO3D 4L`. Consistent with Li et al. (2020b), we observe the performance saturating at four layers. Finally, the autoregressive `TFNO2D` performs remarkably well across all parameter ranges—saturating at an unparalleled RMSE score of $4 \times 10^{-3}$—while, at the same time, constituting a reasonable trade-off between memory consumption and runtime complexity (see Figure 8 in Appendix A.2.1). From this we conclude that, at least for periodic fluid flow simulation, when one is not interested in neural scaling, `FNO2D` marks a promising choice, suggesting its application to real-world weather forecasting scenarios.

### 3.1.2 EXPERIMENT 2: LARGE REYNOLDS NUMBER, 1 K SAMPLES

In this experiment, we evaluate the consistency of the model order found in experiment 1. To do so, we generate more turbulent data by increasing the Reynolds Number $Re$ by an order of magnitude, yielding $Re = 1 \times 10^4$, and reducing the simulation time and sequence length to $T = 30$ timesteps. With an interest in the performance of intrinsically stable models, we discard architectures that depend on gradient clipping and make the same observations as in experiment 1. `TFNO2D` is confirmed as the most accurate model, followed by `SwinTransformer`, `TFNO3D`, and `U-Net` on this harder task. See Figure 1 (right) and Figure 10 in Appendix A.2.2 for quantitative and qualitative results.

---

[6]We observe a similar behavior (not shown) when experimenting with the number of blocks vs. layers in `SwinTransformer`, suggesting to prioritise more layers per block over more blocks with less layers.

### 3.1.3 Experiment 3: Large Reynolds Number, 10 k samples

In this experiment, we aim to understand whether our conclusions still hold when increasing the dataset size. Note that in experiment 2, the three-dimensional TFNO models with `#params` $\geq 8\,\mathrm{M}$ start to show a tendency to overfit (see Figure 12 in Appendix A.2.2). We repeat this experiment and increase the number of training samples by an order of magnitude to $10\,\mathrm{k}$, while reducing the number of epochs from 500 to 50 to preserve the same number of weight updates. Figure 9, Figure 11 and Table 4 in Appendix A.2.2 show that the same findings hold in experiment 3, where `TFNO2D` is affirmed as the most accurate model, followed by `SwinTransformer`, `TFNO3D`, and `U-Net`.

## 3.2 Real-World Weather Data

We extend our analysis to real-world data from WeatherBench (Rasp et al., 2020). Our goal is to evaluate the transferability of the results obtained in Section 3.1 on synthetic data to a more realistic setting. In particular, we seek to provide answers to the following three questions:

(1) Which DLWP model and backbone are most suitable for short- to mid-ranged weather forecasting out to 14 days, according to RMSE and anomaly correlation coefficient (ACC) metrics? (Section 3.2.1)

(2) How stable and reliable are the different methods for long-ranged rollouts when generating predictions out to 365 days and far beyond? (Section 3.2.2)

(3) To what degree do different models adhere to physics and meteorological phenomena by generating forecasts that exhibit characteristic zonal wind patterns? (Section 3.2.3)

Additionally, with respect to these questions, we investigate the role of data representation by either training models on the equirectangular latitude-longitude (LatLon) grid, as provided by ERA5, or on the HEALPix (HPX) mesh (Gorski et al., 2005), which separates the sphere into twelve faces, effectively dissolving data distortions towards the poles.

**Data Selection**  In order to reduce the problem's computational complexity and following earlier DLWP research (Weyn et al., 2020; Karlbauer et al., 2023), we choose a set of 8 expressive core variables on selected pressure levels among the 17 prognostic variables in WeatherBench. Our selection includes four constant inputs in the form of latitude and longitude coordinates, topography, and a land-sea mask. As forcing, we provide the models with precomputed top-of-atmosphere incident solar radiation as input, which is not the target for prediction. Lastly, a set of 8 prognostic variables spans from air temperature at $2\,\mathrm{m}$ above ground ($T_{2m}$) and at a constant pressure level of $850\,\mathrm{hPa}$ ($T_{850}$), to $u$- and $v$-wind components $10\,\mathrm{m}$ above ground (i.e., east-to-west and north-to-south, referred to as zonal $U_{10m}$ and meridional $V_{10m}$ winds, respectively), to geopotential[7] at the four pressure levels 1000, 700, 500, and $300\,\mathrm{hPa}$ (e.g., $\Phi_{500}$). We choose a resolution of $5.625\,°$, which translates to $64 \times 32$ pixels, and operate on a time delta of $\Delta t = 6\,\mathrm{h}$, following common practice in DLWP research.

**Model Setup**  We vary the parameter counts of all models in the range of $50\,\mathrm{k}$, $500\,\mathrm{k}$, $1\,\mathrm{M}$, $2\,\mathrm{M}$, $4\,\mathrm{M}$, $8\,\mathrm{M}$, $16\,\mathrm{M}$, $32\,\mathrm{M}$, $64\,\mathrm{M}$, and $128\,\mathrm{M}$, where the two largest counts are only applied to selected models that did not saturate on fewer parameters. See Table 5 in Appendix B.1 for details about the specific architecture modifications to obtain the respective parameter counts. In summary, our benchmark consists of 179 models—each trained three times, yielding 537 models in total—allowing for a rigorous comparison of DLWP models under controlled conditions on a real-world dataset.

**Optimization**  To prevent predictions from regressing to the mean—where models approach climatology with increasing lead time by generating smooth and blurry outputs—we follow Karlbauer et al. (2023) and constrain the optimization cycle to $24\,\mathrm{h}$, resulting in four autoregressive model calls during training. That is, after receiving the initial condition at time 00:00, the models iteratively unroll predictions for 06:00, 12:00, 18:00, and 24:00. All models are trained on data from 1979 through 2014, evaluated on data from 2015-2016, and tested on the period from 2017 to 2018. We

---

[7]Geopotential, denoted as $\Phi$ with unit $m^2 s^{-2}$, differs from geopotential height, denoted as $Z = \Phi/g$ with unit $m$, where $g = 9.81\,\mathrm{ms}^{-2}$ denotes standard gravity.

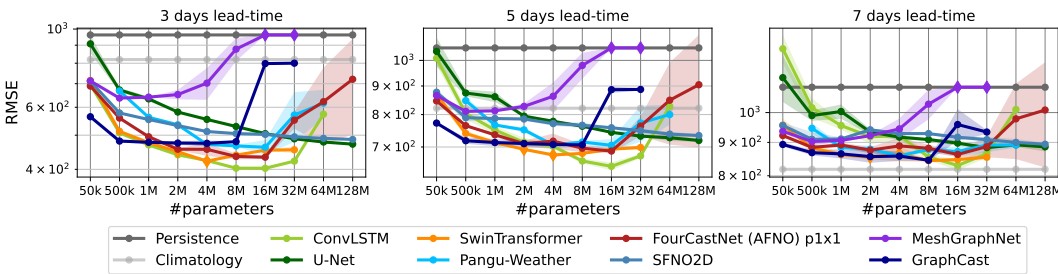

Figure 2: RMSE scores on $\Phi_{500}$ (geopotential at a height of $500\,\mathrm{hPa}$ atmospheric pressure) at three different lead times (3 days left, 5 days center, 7 days right) vs. the number of parameters for DLWP models and backbones trained on a selected set of variables from the WeatherBench dataset.

train each model for 30 epochs with three different random seeds to capture outliers, at least to a minimal degree, using gradient-clipping (by norm) and an initial learning rate of $\eta = 1 \times 10^{-3}$ (unless specified differently) that decays to zero according to a cosine scheduling.

**Evaluation** Typically, DLWP models are evaluated on two leading metrics, i.e., RMSE and ACC, which we also use in our study. The $\mathrm{ACC} \in [-1, 1]$ denotes how well the model captures anomalies in the data. A forecast is called skillful in the range $1.0 \leq \mathrm{ACC} \leq 0.6$, whereas an $\mathrm{ACC} < 0.6$ is considered imprecise and useless. For long-ranged rollouts, different methods find application, e.g., qualitatively inspecting the raw output fields at long lead times (Weyn et al., 2021; Bonev et al., 2023), comparing spatial spectra of model outputs (Karlbauer et al., 2023; McCabe et al., 2023), or computing averages over time periods of months, years, or more (Watt-Meyer et al., 2023). We inspect the soundness of raw output fields visually and quantitatively compare monthly averages for assessing performance at long lead times as well as by computing spectra.

### 3.2.1 Short- to Mid-Ranged Forecasts

Useful weather forecasts (called 'skillful' in meteorological terms) can be expected on lead times out to at most 14 days (Bauer et al., 2015). Afterwards, the chaotic nature of the planet's atmosphere prevents the determination of an accurate estimate of weather dynamics (Lorenz, 1963; Palmer et al., 2014). We quantify and compare the forecast quality of the benchmarked DLWP models for lead times of 3, 5, and 7 days via RMSE and ACC scores to assess how different models perform on lead times that are relevant for end users on a daily basis.

Our evaluation of $\Phi_{500}$ forecasts at lead times up to seven days reveals a consistent reduction of forecast error when increasing the number of parameters across models, as shown as point-wise results at three, five, and seven days lead time in Figure 2. The scaling behavior[8] differs substantially between models, featuring `U-Net` to stand out as the only model that keeps improving monotonically with more parameters. In contrast, all other models exhibit a point, individually

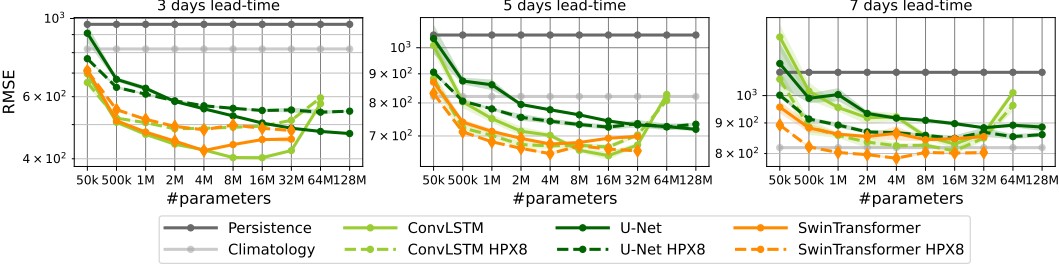

Figure 3: RMSE on $\Phi_{500}$ for different models trained on the LatLon (solid lines) or on the HEALPix (HPX, dashed lines) mesh. When operating on the distortion-reducing HEALPix mesh, all three benchmarked methods improve their forecast performance at longer lead times.

---

[8]Not "neural scaling" behavior, as we do not observe that, to be clear.

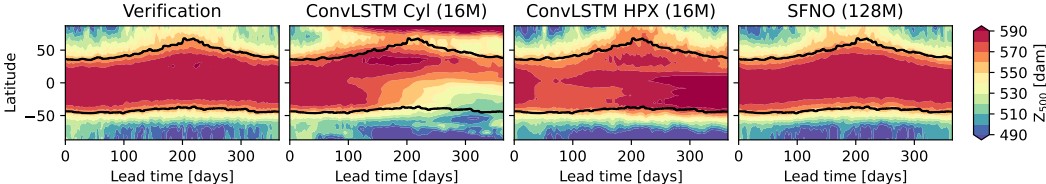

Figure 4: Zonally averaged $Z_{500}$ (geopotential height at an atmospheric pressure of $500\,\text{hPa}$) forecasts of selected models initialized on Jan. 01, 2017, and run forward for 365 days. The verification panel (left) illustrates the seasonal cycle, where lower air pressures are observed on the northern hemisphere in Jan., Feb., Nov., Dec., and higher pressures in Jul., Aug., Sep. (and vice versa on the southern hemisphere). The black line indicates the $540\,\text{dem}$ (in decameters) progress and is added to each panel to showcase how each model's forecast captures the seasonal trend.

differing for each architecture, beyond which a further increase of parameters leads to an increase in forecast error, deteriorating model performance. Beyond this parameter count, the models no longer exhibit converging training curves, but stall at a constant error level. This demonstrates difficulties in optimizing the models when having more degrees of freedom, which lead to more complex error landscapes with more local minima where the algorithm can get stuck (Geiger et al., 2021; Krishnapriyan et al., 2021). Intriguingly, the recurrent `ConvLSTM` with $16\,\text{M}$ parameters yields accurate predictions on short lead times, even though it is trained and tested on sequence lengths of 4 and 56, respectively. It eventually falls behind the other models at a lead time of seven days. While `SwinTransformer` and `FourCastNet` challenge `ConvLSTM` on their best parameter counts, `GraphCast` is superior in the low-parameter regime albeit exhibiting less improvements with more parameters. Interestingly, we observe `Pangu-Weather` scoring worse than the backbone it is based on, namely `SwinTransformer`, at least in short- to mid-ranged horizons.[9] Due to the unexpectedly[10] good performance of `FourCastNet` and poor results for Spherical FNO (`SFNO`), we explore and contrast these architectures, along with their (T)FNO backbones, more rigorously in Appendix B.3. Additional results on air temperature and other target variables (cf. Figure 20 and following in Appendix B.4), demonstrate similar trends and model rankings (`SFNO` ranking higher) across target variables on RMSE and also on anomaly correlation coefficient (ACC) metrics.

To investigate the role of data representations, i.e., differentiating between a naïve rectangular and a sophisticated spherical grid, we project the LatLon data to the HEALPix mesh and modify `ConvLSTM`, `U-Net`, and `SwinTransformer` accordingly to train them on the distortion-reduced mesh. In Figure 3, we observe that all models benefit from the data preprocessing, likely due to reduced data distortions, which relieves the models from having to learn a correction of area with respect to latitude. Improvements are consistent across architecture and parameter count, being more evident on larger lead times. Given that the HEALPix mesh used here only counts $8 \times 8 \times 12 = 768$ pixels, the improvement over the LatLon mesh with $64 \times 32 = 2048$ pixels is even more significant. This underlines the benefit of explicit spherical data representations, which also find applications in sophisticated DLWP models, e.g., `Pangu-Weather`, `SFNO`, and `GraphCast`.

### 3.2.2 LONG-RANGED ROLLOUTS

The stability of weather models is key for long-range projections on climate scales. We investigate the stability of the trained DLWP models by running them in a closed loop out to 365 days. Models that produce realistic states on that horizon—which we assess by inspecting the divergence from monthly averaged $\Phi_{500}$ predictions—are considered promising starting points for model development on climate scales.

We evaluate the suitability of models for long-range predictions in two ways. First, we inspect the state produced by selected models at a lead time of 365 days. This provides the first insights into the stability of different models, where only a subset of models produces an appealing realization of the $Z_{500}$ field.

---

[9]We cannot guarantee that we optimized each model in the most suitable way for the respective architecture. An exhaustive exploration of hyperparameters for each model—beyond a directed search when our results did not match with those in the literature—would be nearly intractable.

[10]Compared to Bonev et al. (2023), where SFNO is reported to outperform FourCastNet at five-days lead time.

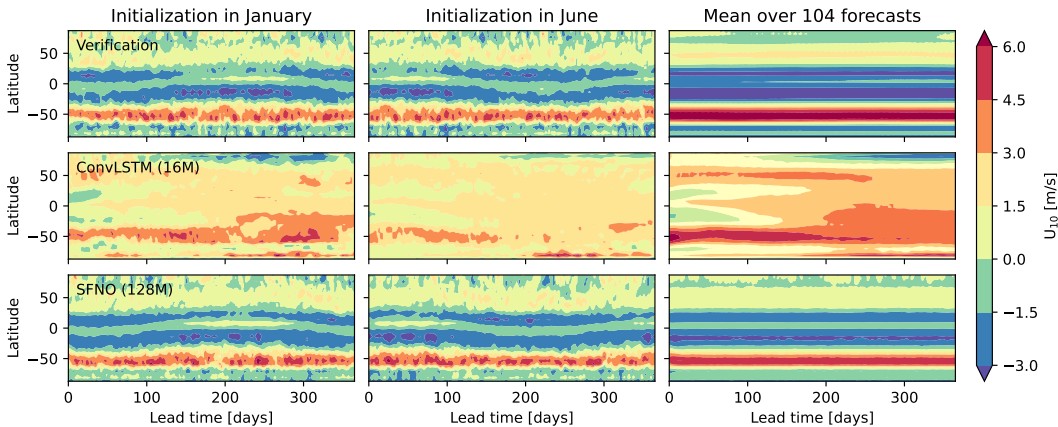

Figure 5: Zonally averaged $U_{10}$ winds over 365 days lead time displayed for verification (first row), `ConvLSTM` with $16\,\text{M}$ parameters (second row), and `SFNO` with $128\,\text{M}$ parameters (third row). Left and center showcase single rollouts initialized in January and June, respectively, while the right-most panel provides an average computed over all 104 forecasts, initialized from January through December 2017. While `SFNO` (third row) neatly reproduces the annual distribution of winds, showing the importance of spherical representation, `ConvLSTM` (second row) fails at capturing these dynamics on long forecast ranges.

This subset includes `SwinTransformer HPX` (on the HEALPix mesh), `FourCastNet` with different patch sizes, `SFNO`, `Pangu-Weather`, and `GraphCast` (see Figure 13 in Appendix B.2). Other methods blow up and disqualify for long-ranged forecasts. Second, zonally averaged predictions of $Z_{500}$ over 365 days in the forecasts (see Figure 4) indicate points in time where the models blow up if they do. For example, `ConvLSTM Cyl` (on the cylinder mesh) predicts implausibly high pressures in high latitudes near the north pole already after a few days, whereas `ConvLSTM HPX` begins to loose the high pressure signature in the tropics after 40 days into the forecast. See Figure 14 in Appendix B.2 for more examples.

To expand beyond one year, we run selected models out to 50 years and observe a similar behavior, supporting `SwinTransformer`, `FourCastNet`, `SFNO`, `Pangu-Weather`, and `GraphCast` as stable models (see Appendix B.2 and Figure 16 for details and Figure 17 for power spectra).

### 3.2.3 PHYSICAL SOUNDNESS

Here, we seek to elucidate whether and to which degree the models replicate physical processes. To this end, we compare how each model generates zonal surface wind patterns, known as Trade Winds (or Easterlies) and Westerlies. Easterlies (west-to-east propagating winds) are pronounced in the tropics, from 0 to 30 degrees north and south of the equator, whereas Westerlies (east-to-west propagating winds) appear in the extratropics of both hemispheres at around 30 to $60\,^\circ$. Westerlies are more emphasized in the southern hemisphere, where the winds are not slowed down as much by land masses. For visualizations and details about global wind patterns and circulations, see encyclopedias for atmospheric sciences.[11][12] Figure 5 illustrates these winds when observed in the individual forecasts of `ConvLSTM` (second row) and `SFNO` (third row) and compared to the verification (first row). When averaging over the entire lead time out to 365 days and over 104 forecasts (initialized bi-weekly from January through December 2017), the wind patterns are shown clearly and we investigate how accurately each model reproduces these patterns. `SFNO` most accurately generates Easterlies and Trade Winds, likely due to its physically motivated inductive bias in the form of spherical harmonics. This allows SFNO to adhere to physical principles, whereas `ConvLSTM` misses such an inductive bias, resulting in physically implausible predictions on longer lead times.

---

[11]http://ww2010.atmos.uiuc.edu/(Gh)/wwhlpr/global_winds.rxml.
[12]https://www.eoas.ubc.ca/courses/atsc113/sailing/met_concepts/
09-met-winds/9a-global-wind-circulations/.

We complement these results by quantitative RMSE scores in Figure 15 in Appendix B.2. Most prominently, `SFNO`, `FourCastNet` (featuring $1 \times 1$ patches), and `Pangu-Weather` reliably exhibit the wind patterns of interest, mostly achieving errors below `Persistence`. Other methods either score worse than `Persistence` or even exceed an error threshold of $100\,\text{m/s}$. Models exceeding this threshold are discarded from the plot and considered inappropriate—given `Persistence` produces an RMSE of $1.16$, $1.41$, and $1.56\,\text{m/s}$ for Trade Winds, South Westerlies, and global wind averages, respectively.

## 4 DISCUSSION

In this work, we obtain insights into which DLWP models are more suitable for weather forecasting by devising controlled experiments. In particular, we fix the input data and training protocol, and we vary the architecture and number of parameters. First, in a limited setup on synthetic periodic Navier-Stokes data, we find that `TFNO2D` performs the best at predicting the dynamics, followed by `TFNO3D`, `SwinTransformer`, `FNO3D`, `ConvLSTM`, `U-Net`, `FourCastNet`, and `MS MeshGraphNet`. Although we enable circular padding in the compared architectures, the periodic nature of the Navier-Stokes data likely favors the inductive bias of FNO. Second, when extending our analysis to real-world data, we observe that FNO backbones fall behind `ConvLSTM`, `SwinTransformer`, and `FourCastNet` on lead times up to 14 days. We attribute this drop in accuracy of FNO to the non-periodic equirectangularily represented WeatherBench data, which connects to the finding in Saad et al. (2023) that FNO does not satisfy boundary conditions. On lead times out to 365 days, `SFNO`, `Pangu-Weather`, and `GraphCast` generate physically adequate outputs. This encourages the implementation of appropriate inductive biases—e.g., periodicity in FNO for Navier-Stokes, spherical representation in SFNO, or the HEALPix mesh on WeatherBench—to facilitate stable model rollouts. In our experiments, `GraphCast` outperforms other methods in the small parameter regime, but it does not keep up with other models when increasing the parameter count. This underlines `GraphCast`'s potential, but it also highlights the challenges of training graph-based methods.

Our results also show that all methods (with accompanying training protocols, etc.) saturate or deteriorate (with increasing parameters, data, or compute), demonstrating that further work is needed to understand the possibilities of neural scaling in these (and other) classes of scientific machine learning models. From an applicability viewpoint, our results provide insights into the ease or difficulties, potentially arising during model training, that users should be aware of when choosing a respective architecture. We sparingly explore hyperparameters in selected cases on WeatherBench, where our results deviate substantially from the literature, i.e., for `GraphCast`, `SFNO`, and `FourCastNet`.

In summary, our results suggest the consideration of `ConvLSTM` blocks when aiming for short-to-mid-ranged forecasts. Due to the recurrent nature of `ConvLSTM` cells, these models may benefit from longer training horizons—i.e., sequence lengths beyond the four prediction steps intentionally used for the deterministic models in this work. This stands in conflict with the phenomenon of approaching climatology when training on longer lead times. We also find `SwinTransformer` to be an accurate model that is amenable to straightforward training. It is a more expensive model, though, in terms of memory and inference time (see Figure 23 in Appendix B.4 for a thorough runtime and memory comparison). For long lead times, the sophisticated designs of `SFNO`, `FourCastNet`, `Pangu-Weather`, and `GraphCast` prove to be advantageous. The design of recurrent probabilistic DLWP models (that provide an uncertainty estimation as output) is a promising direction for future research (Gao et al., 2023; Cachay et al., 2023; Price et al., 2023) as well as the incorporation of established physical relations such as conservation laws (Hansen et al., 2023).

In our repository, we provide model checkpoints and selected model output files and encourage researchers to conduct further analyses. Additionally, our training protocol can be adapted to include more input variables or to operate on finer resolutions, as provided through WeatherBench.

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

# A    NAVIER-STOKES EXPERIMENTS

## A.1    MODEL, DATA, AND TRAINING SPECIFICATIONS

In this section, we discuss the model configurations and how we vary the number of parameters in our experiments. In addition, we detail the dataset generation and training protocols.

### A.1.1    MODEL CONFIGURATIONS

We compare six model classes that form the basis for SOTA DLWP models. We provide details about each model and how we modify them in order to vary the number of parameters below. Table 2 provides an overview and summary of the parameters and model configurations.

**ConvLSTM**    We first implement an encoder—to increase the model's receptive field—consisting of three convolutions with kernel size $k = 3$, stride $s = 1$, padding $p = 1$, set padding_mode $=$ circular to match the periodic nature of our data, and implement $\tanh$ activation

Table 2: Model configurations partitioned by model and number of parameters (which amount to the trainable weights). For configurations that are not specified here, the default settings from the respective model config files are applied, e.g., `ConvLSTM` employs the default from `configs/model/convlstm.yaml`, while overriding `hidden_sizes` by the content of the "Dim." column of this table. Details are also reported in the respective model paragraphs of Appendix A.1.1.

| Model | #params | Enc. | Dim. | Dec. | Model | Dim. (hidden sizes) |
|---|---|---|---|---|---|---|
| `ConvLSTM` | 5 k | | $4 \times 4$ | | `U-Net` | $[1, 2, 4, 8, 8]$ |
| | 50 k | | $4 \times 13$ | | | $[3, 6, 12, 24, 48]$ |
| | 500 k | $3 \times$ Conv2D with $\tanh()$ | $4 \times 40$ | $1 \times$ Conv2D | | $[8, 16, 32, 64, 128]$ |
| | 1 M | | $4 \times 57$ | | | $[12, 24, 48, 96, 192]$ |
| | 2 M | | $4 \times 81$ | | | $[16, 32, 64, 128, 256]$ |
| | 4 M | | $4 \times 114$ | | | $[23, 46, 92, 184, 368]$ |
| | 8 M | — | — | — | | $[33, 66, 132, 264, 528]$ |

| Model | #params | #modes | Dim. | #layers | Model | #modes | Dim. | #layers |
|---|---|---|---|---|---|---|---|---|
| `(T)FNO3D L1-16` | 5 k | $3 \times 3$ | 11 | 1 | `TFNO2D L4` | $2 \times 12$ | 2 | 4 |
| | 50 k | $3 \times 3$ | 32 | 1 | | $2 \times 12$ | 8 | 4 |
| | 500 k | $3 \times 7$ | 32 | 1 | | $2 \times 12$ | 27 | 4 |
| | 1 M | $3 \times 10$ | 32 | 1 | | $2 \times 12$ | 38 | 4 |
| | 2 M | $3 \times 12$ | 32 | 1 | | $2 \times 12$ | 54 | 4 |
| | 4 M | $3 \times 12$ | 32 | 2 | | $2 \times 12$ | 77 | 4 |
| | 8 M | $3 \times 12$ | 32 | 4 | | $2 \times 12$ | 108 | 4 |
| | 16 M | $3 \times 12$ | 32 | 8 | | $2 \times 12$ | 154 | 4 |
| | 32 M | $3 \times 12$ | 32 | 16 | | — | — | — |

| Model | #params | #modes | Dim. | #layers | Model | Dim. | #layers |
|---|---|---|---|---|---|---|---|
| `TFNO3D L4` | 5 k | — | — | — | `FourCastNet` | 12 | 1 |
| | 50 k | $3 \times 12$ | 2 | 4 | | 64 | 1 |
| | 500 k | $3 \times 12$ | 8 | 4 | | 112 | 4 |
| | 1 M | $3 \times 12$ | 11 | 4 | | 160 | 4 |
| | 2 M | $3 \times 12$ | 16 | 4 | | 232 | 4 |
| | 4 M | $3 \times 12$ | 22 | 4 | | 326 | 4 |
| | 8 M | $3 \times 12$ | 32 | 4 | | 468 | 4 |

| Model | #params | #heads | Dim. | #blocks | #lrs/blck | Model | $D_{\mathrm{processor}}$ | $D_{\mathrm{other}}$ |
|---|---|---|---|---|---|---|---|---|
| `Swin-Transformer` | 5 k | 1 | 8 | 1 | 1 | `MS Mesh-GraphNet` | 8 | 8 |
| | 50 k | 2 | 8 | 2 | 2 | | 34 | 32 |
| | 500 k | 4 | 40 | 2 | 4 | | 116 | 32 |
| | 1 M | 4 | 60 | 2 | 4 | | — | — |
| | 2 M | 4 | 88 | 2 | 4 | | — | — |

functions. We add four `ConvLSTM` cells, also with circular padding and varying channel depth (see Table 2 for details), followed by a linear output layer. Being the only recurrent model, we perform ten steps of teacher forcing before switching to closed loop to autoregressively unroll a prediction into the future.

**U-Net**   We implement a five-layer encoder-decoder architecture with avgpool and transposed convolution operations for down and up-sampling, respectively. On each layer, we employ two consecutive convolutions with ReLU activations (Fukushima, 1975) and apply the same parameters described above in the encoder for `ConvLSTM`. See Table 2 for the numbers of channels hyperparameter setting.

**SwinTransformer**   Enabling circular padding and setting patch size $p = 2$, we benchmark the shifted window transformer (Liu et al., 2021) by varying the number of channels, heads, layers, and blocks, as detailed in Table 2, while keeping remaining parameters at their defaults.

**MS MeshGraphNet**   We formulate a periodically connected graph to apply Multi-Scale Mesh-GraphNet (`MS MeshGraphNet`) with two stages, featuring 1-hop and 2-hop neighborhoods, and follow Fortunato et al. (2022) by encoding the distance and angle to neighbors in the edges. We employ four processor and two node/edge encoding and decoding layers and set $\text{hidden\_dim} = 32$ for processor, node encoder, and edge encoder, unless overridden (see Table 2).

**FNO**   We compare three variants of `FNO`: Two three-dimensional formulations, which process the temporal and both spatial dimensions simultaneously to generate a three-dimensional output of shape $[T, H, W]$ in one call, and a two-dimensional version, which only operates on the spatial dimensions of the input and autoregressively unrolls a prediction into the future. While fixing the lifting and projection channels at 256, we vary the number of Fourier modes, channel depth, and number of layers according to Table 2.

**FourCastNet**   We choose a patch size of $p = 4$, fix $\text{num\_blocks} = 4$, enable periodic padding in both spatial dimensions, and keep the remaining parameters at their default values while varying the number of layers and channels as specified in Table 2.

### A.1.2   DATA GENERATION

We provide additional information about the data generation process in Table 3, which we keep as close as possible to that reported in Li et al. (2020b) and Gupta et al. (2021).

### A.1.3   TRAINING PROTOCOL

In the experiments, we use the Adam optimizer with learning rate $\eta = 1 \times 10^{-3}$ (except for `MS MeshGraphNet`, which only converged with a smaller learning rate of $\eta = 1 \times 10^{-4}$) and cosine learning rate scheduling to train all models with a batch size of $B = 4$, effectively realizing $125\,\text{k}$

Table 3: Settings for training, validation, and test data generation in the experiments, where $f$, $T$, $\delta_t$, and $\nu$ denote the dynamic forcing, sequence length (corresponding to the simulation time, which, in our case, matches the number of frames, i.e., $\Delta t = 1$), time step size for the simulation, and viscosity (which is the inverse of the Reynolds Number, i.e., $Re = 1/\nu$), respectively. The parameters $\alpha$ and $\tau$ parameterize the Gaussian random field to sample an initial condition (IC) resembling the first timestep.

| | | | Simulation parameters | | IC | | #samples | | |
| Experiment | $f$ | $T$ | $\delta_t$ | $\nu$ | $\alpha$ | $\tau$ | Train | Val. | Test |
|---|---|---|---|---|---|---|---|---|---|
| 1 | * | 50 | $1 \times 10^{-2}$ | $1 \times 10^{-3}$ | 2.5 | 7 | 1000 | 50 | 200 |
| 2 | * | 30 | $1 \times 10^{-4}$ | $1 \times 10^{-4}$ | 2.5 | 7 | 1000 | 50 | 200 |
| 3 | * | 30 | $1 \times 10^{-4}$ | $1 \times 10^{-4}$ | 2.5 | 7 | 10000 | 50 | 200 |

$^*f = 0.1(\sin(2\pi(x + y)) + \cos(2\pi(x + y)))$, with $x, y \in [0, 1, \ldots, 63]$.

weight update steps, relating to 500 and 50 epochs, respectively, for $1\,\mathrm{k}$ and $10\,\mathrm{k}$ samples.[13] For the training objective and loss function, we choose the mean squared error (MSE) between the model outputs and respective ground truth frames, that is $\mathcal{L} = \mathrm{MSE}(\hat{y}_{h+1:T}, y_{h+1:T})$. Note that, to stabilize training, we have to employ gradient clipping (by norm) for selected models, indicated by italic numbers in tables and triangle markers in figures.

## A.2 ADDITIONAL RESULTS AND MATERIALS

In this section, we provide additional empirical results for the three experiments on Navier-Stokes dynamics.

### A.2.1 RESULTS FROM EXPERIMENT 1: LARGE REYNOLDS NUMBER, 1 K SAMPLES

Figure 6 illustrates the initial and end conditions along with the respective predictions of all models. Qualitatively, we find there exist parameter settings for all models to successfully unroll a plausible prediction of the Navier-Stokes dynamics over 40 frames into the future, as showcased by the last predicted frame, i.e., $\hat{y}_{t=T}$ (see the third and fifth row of Figure 6). When computing the difference between the prediction and ground truth, i.e., $d = \hat{y} - y$, we observe clear variations in the accuracy of the model outputs, denoted by the saturation of the difference plots in the second and fourth row of Figure 6. Interestingly, this difference plot also reveals artifacts in the outputs of selected models: `SwinTransformer` and `FourCastNet` generate undesired patterns that resemble their windowing and patching mechanisms, whereas the 2-hop neighborhood, which was chosen as the resolution of the coarser grid, is baked into the output of `MS MeshGraphNet`. According to

Table 4: RMSE scores partitioned by experiments and reported for each model under different numbers of parameters. Errors reported in italic correspond to models that had to be retrained with gradient clipping (by norm) due to stability issues. With `OOM` and `sat`, we denote models that ran out of GPU memory and saturated, meaning that we did not train models with more parameters because the performance already saturated over smaller parameter ranges. Best results are shown in bold. More details about architecture specifications are reported in Appendix A.1.1 and Table 2.

| | Model | #params | | | | | | | | |
| | | 5 k | 50 k | 500 k | 1 M | 2 M | 4 M | 8 M | 16 M | 32 M |
|---|---|---|---|---|---|---|---|---|---|---|
| **Experiment 1** | Persistence | .5993 | .5993 | .5993 | .5993 | .5993 | .5993 | .5993 | .5993 | .5993 |
| | ConvLSTM | .1278 | .0319 | .0102 | *.0090* | *.2329* | *.4443* | OOM | —- | —- |
| | U-Net | .5993 | .0269 | .0157 | .0145 | .0131 | .0126 | .0126 | sat | —- |
| | FNO3D L1-8 | .3650 | .2159 | .1125 | .1035 | .1050 | .0383 | .0144 | .0095 | —- |
| | TFNO3D L1-16 | —- | —- | —- | .0873 | .0889 | .0221 | .0083 | .0066 | .0069 |
| | TFNO3D L4 | —- | .0998 | .0173 | .0127 | .0107 | .0091 | .0083 | sat | —- |
| | TFNO2D L4 | **.0632** | **.0139** | **.0055** | **.0046** | **.0043** | **.0054** | **.0041** | **.0046** | sat |
| | SwinTransformer | .1637 | .0603 | .0107 | .0084 | .0070 | OOM | —- | —- | —- |
| | FourCastNet | .1558 | .0404 | .0201 | .0154 | *.0164* | *.0153* | *.0149* | sat | —- |
| | MS MeshGraphNet | *.2559* | *.0976* | *.5209* | OOM | —- | —- | —- | —- | —- |
| **Experiment 2** | Persistence | 1.202 | 1.202 | 1.202 | 1.202 | 1.202 | 1.202 | 1.202 | 1.202 | 1.202 |
| | U-Net | —- | .3874 | .3217 | .3117 | .3239 | .3085 | sat | —- | —- |
| | TFNO3D L1-8 | —- | —- | —- | —- | .5407 | .3811 | .3105 | .3219 | sat |
| | TFNO3D L4 | —- | .5038 | .3444 | .3261 | .3224 | .3155 | .3105 | sat | —- |
| | TFNO2D L4 | **.4955** | **.3091** | **.2322** | **.2322** | **.2236** | **.2349** | **.2358** | sat | —- |
| | SwinTransformer | .6266 | .4799 | .2678 | .2552 | .2518 | OOM | —- | —- | —- |
| **Experiment 3** | Persistence | 1.202 | 1.202 | 1.202 | 1.202 | 1.202 | 1.202 | 1.202 | 1.202 | 1.202 |
| | U-Net | —- | .3837 | .3681 | .2497 | .3162 | .2350 | .2383 | sat | —- |
| | TFNO3D L1-16 | —- | —- | —- | —- | .5146 | .2805 | .1814 | .1570 | .1709 |
| | TFNO3D L4 | —- | .4799 | .2754 | .2438 | .2197 | .2028 | .1814 | .1740 | sat |
| | TFNO2D L4 | **.4846** | **.2897** | **.1778** | **.1585** | **.1449** | **.1322** | **.1248** | **.1210** | sat |
| | SwinTransformer | .6187 | .4698 | .2374 | .2078 | .1910 | OOM | —- | —- | —- |

---

[13] With an exception for `MS MeshGraphNet`, which only supports a batch size of $B = 1$, resulting in $500\,\mathrm{k}$ weight update steps.

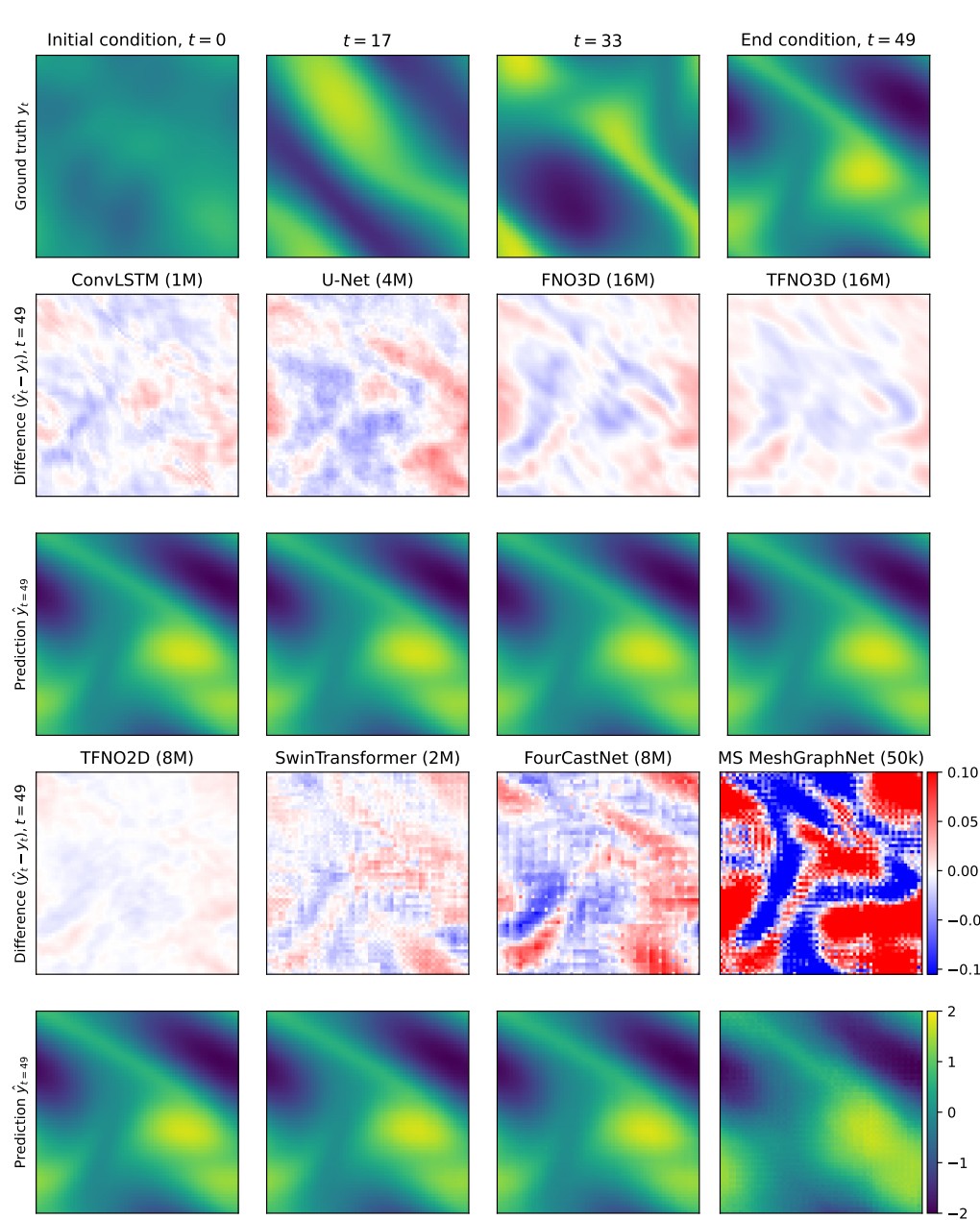

Figure 6: Qualitative results on the Navier-Stokes dataset with Reynolds Number $Re = 1 \times 10^3$ trained on $1\,\mathrm{k}$ samples (experiment 1). The first row shows the ground truth at four different points in time. The remaining rows show the difference between the predicted- and ground-truth at final time (row two and four), as well as the predicted final frame (row three and five). All models receive the first 10 frames of the sequence to predict the remaining 40 frames. The last frame of the predicted sequence from the best models are visualized and respective parameter counts are displayed in parenthesis.

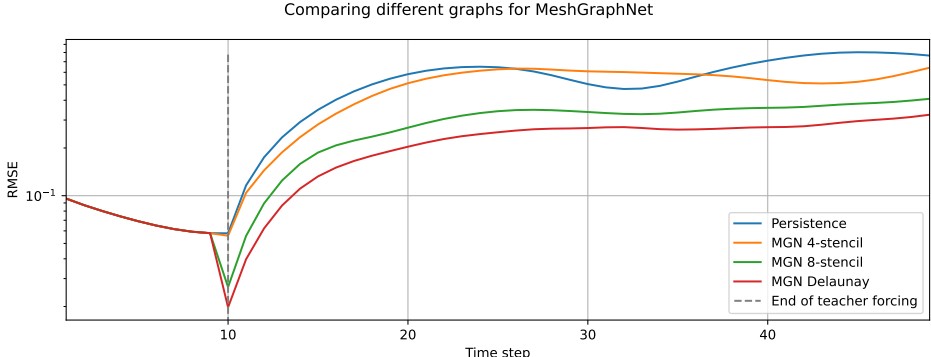

Figure 7: RMSE evolving over forecast time for three different underlying graphs (meshes) that are used in the single scale MeshGraphNet (MGN) (Pfaff et al., 2020).

the lowest error scores reported in Table 4, we only visualize the best performing model among all parameter ranges in Figure 6 and observe the trend that `TFNO2D` performs best, followed by `TFNO3D`, `SwinTransformer`, `FNO3D`, `ConvLSTM`, `U-Net`, `FourCastNet`, and `MS MeshGraphNet`.

Next, we study the effect of the underlying graph in GNNs. Observing the poor behavior of `MS MeshGraphNet` in Figure 6, we investigate the effect of three different periodic graph designs to represent the neighborhoods in the GNN. First, the 4-stencil graph connects each node's perpendicular four direct neighbors (i.e., north, east, south, and west) in a standard square Cartesian mesh. Second, the 8-stencil graph adds the direct diagonal neighbors to the 4-stencil graph. Third, the Delaunay graph connects all nodes in the graph by means of triangles, resulting in a hybrid of the 4-stencil and 8-stencil graph, where only some diagonal edges are added. To simplify the problem, we conduct this analysis on the single-scale MeshGraphNet (Pfaff et al., 2020) instead of using the hierarchical `MS MeshGraphNet` (Fortunato et al., 2022). While the graphs have the same number of nodes $|\mathcal{N}| = 4096$, their edge counts differ to $|\mathcal{E}_4| = 16384$, $|\mathcal{E}_8| = 32768$, and $|\mathcal{E}_D| = 24576$ for the 4-stencil, 8-stencil, and Delaunay graph, respectively. The results reported in this paper are based on the 4-stencil graph.

Interestingly, as indicated in Figure 7, the results favor the Delaunay graph over the 8- and 4-stencil graphs, respectively. Apparently, the increased connectedness is beneficial for the task. At the same time, though, the irregularity introduced by the Delaunay triangulation potentially forces the model to

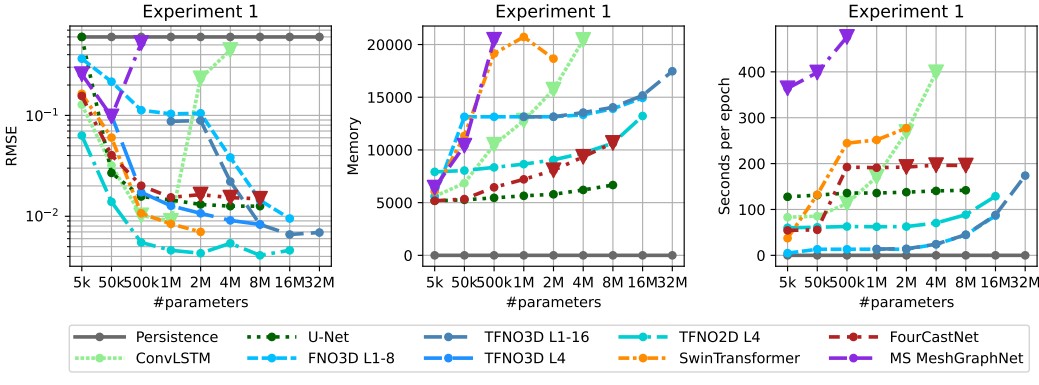

Figure 8: RMSE (left), memory consumption (center), and runtime complexity in seconds per epoch (right) over different parameter counts for models trained on Reynolds Number $Re = 1 \times 10^3$ with $1\,\mathrm{k}$ samples for experiment 1. In Figure 23, we repeat this analysis more thoroughly on real-world data.

develop more informative codes for the edges to represent direction and distance of neighbors more meaningfully.

Lastly, Figure 8 compares the RMSE, memory consumption and computational cost in seconds per epoch as a function of the number of parameters. We see that `TFNO2D L4` performs the best in terms of the RMSE and also scales well with respect to memory and runtime.

### A.2.2 RESULTS FROM EXPERIMENT 2 AND EXPERIMENT 3: LARGE REYNOLDS NUMBER

Table 4 shows the quantitative error scores of all the experiments (for an easier comparability). We see that the same trend occurs across all three experiments with `TFNO2D` performing the best. Figure 9 illustrates the similar trends of these RMSE results from experiments 2 and 3. Figure 10 and Figure 11 provide the qualitative visualizations for experiments 2 and 3, respectively. Figure 9 (right) and Figure 11 for experiment 3 show that, while all models consistently improve their scores

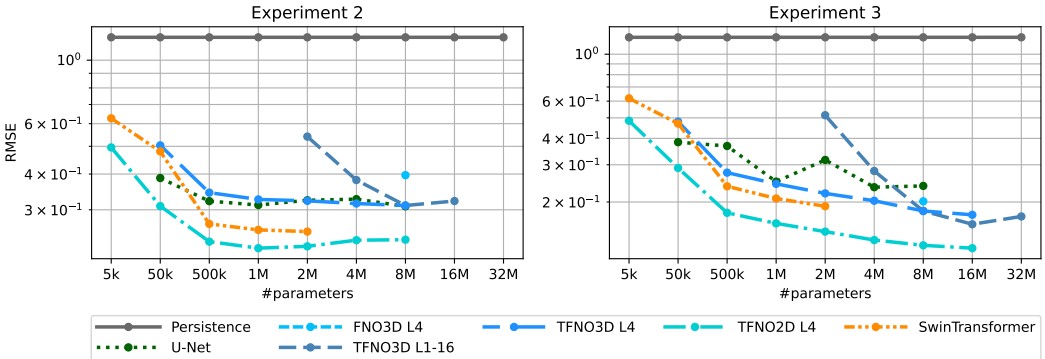

Figure 9: RMSE vs. parameters for models trained on Reynolds Number $Re = 1 \times 10^4$ with $1\,\mathrm{k}$ (experiment 2, left) and $10\,\mathrm{k}$ (experiment 3, right) samples. Note the different y-axis scales. Main observation: As expected, model performance correlates with the number of samples. The number of samples, though, does not affect the model ranking.

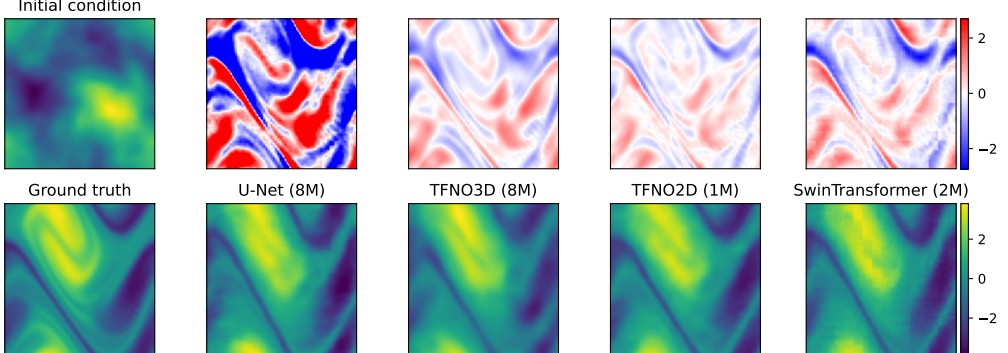

Figure 10: Qualitative results on Navier-Stokes data with Reynolds Number $1 \times 10^4$ trained on $1\,\mathrm{k}$ samples (experiment 2). The top left shows the initial condition. The remaining columns in the top row show the differences between the predicted and ground-truth at the final time for the various models. The bottom left shows the ground truth at the final time. The remaining columns in the bottom row show the final predictions from the various models to visually compare to the ground truth. All models face difficulties at resolving the yellow vortex, resulting in blurry predictions around the turbulent structure at this higher Reynolds Number. Among the parameter ranges, the best models are selected for visualizations (parameter count in brackets).

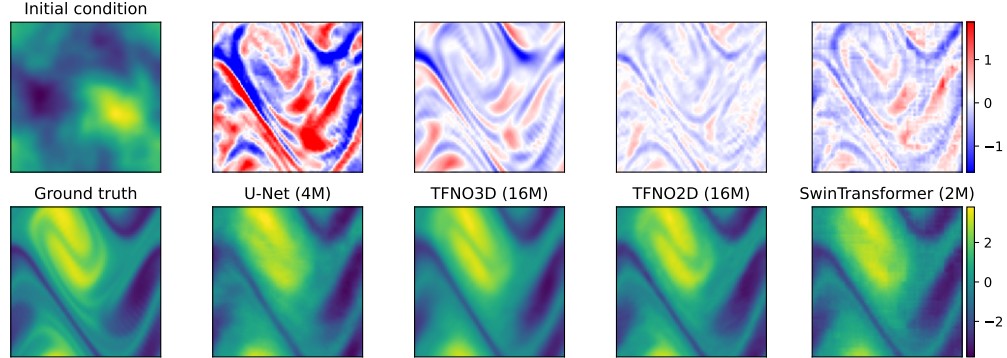

Figure 11: Qualitative results on Navier-Stokes data with Reynolds Number $1 \times 10^4$ trained on $10\,\mathrm{k}$ samples (experiment 3). In comparison to Figure 10, the yellow vortex is captured more accurately by `TFNO2D` as a consequence of the larger training set. See plot description in Figure 10 for details.

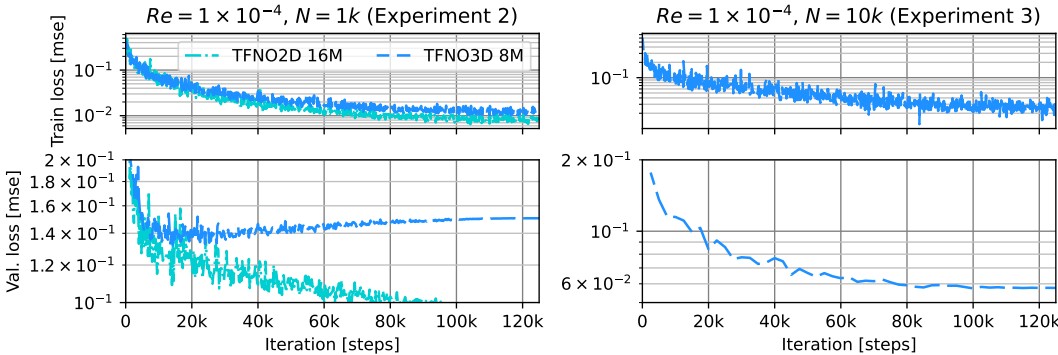

Figure 12: Training (top) and validation (bottom) error curves for `TFNO2D` with $16\,\mathrm{M}$ and `TFNO3D` with $8\,\mathrm{M}$ parameters in experiment 2 and 3 (left and right, respectively). Around iteration $20\,\mathrm{k}$, `TFNO3D` starts to overfit to the training data, as the training error keeps improving, while the validation error stagnates and deteriorates.

due to the larger training set, the results from experiments 1-2 still hold. That is, when comparing the convergence levels in Figure 9 (right) and Table 4, we see that all models saturate at lower error regimes, while the ordering of the model performance from experiment 1 remains unchanged. Figure 10 and Figure 11 illustrate qualitatively that the models benefit from the increase of training samples in experiment 3 since the yellow vortex at this higher Reynolds Number is resolved more accurately when the models are trained on more data. Figure 12, which compares the training and validation curves for `TFNO3D` from both experiments, also shows the benefit of more training data in experiment 3. While the model overfits with $1\,\mathrm{k}$ samples (experiment 2, left), the validation curve does not deteriorate with $10\,\mathrm{k}$ samples (experiment 3, right), which indicates that the increase of training data prevents `TFNO3D` from overfitting. We also see that the two-dimensional `TFNO2D` variant does not overfit in experiment 2.

# B    REAL-WORLD WEATHER DATA

## B.1    MODEL SPECIFICATIONS

In this section, we discuss the model configurations and how we vary the number of parameters in our experiments on WeatherBench.

**ConvLSTM**    Similarly to our experiment on Navier-Stokes data, we implement an encoder consisting of three convolutions with kernel size $k = 3$, stride $s = 1$, padding $p = 1$, set the horizontal padding_mode = circular, and the vertical to zero-padding to match the periodic nature of our data along lines of latitudes, and implement tanh activation functions. We add four ConvLSTM layers, employing the identical padding mechanism and varying channel depth (see Table 5 for details), followed by a linear output layer.

**U-Net**    On rectangular data, we implement a five-layer encoder-decoder architecture with avgpool and transposed convolution operations for down and up-sampling, respectively. When training on HEALPix data, we only employ four layers due to resolution conflicts in the synoptic (bottom-most) layer of the U-Net while controlling for parameters. Irrespective of the mesh, we employ two consecutive convolutions on each layer with ReLU activations (Fukushima, 1975) and apply the same parameters described above in the encoder for ConvLSTM. See Table 5 for the numbers of channels hyperparameter setting.

**SwinTransformer**    Also enabling circular padding along the east-west dimension and setting patch size to $p = 1$, we benchmark the shifted window transformer (Liu et al., 2021) by varying the number of channels, heads, layers, and blocks, as detailed in Table 5, while keeping remaining parameters at their defaults.

**Pangu-Weather**    While based on SwinTransformer, Pangu-Weather implements *earth-specific* transformer layers to inform the model about position on the sphere (via injected latitude-longitude codes) and to be aware of the atmosphere's vertical slicing on respective three-dimensional variables. Since we do not provide fine-grained vertical information across different input channels, we only employ the 2D earth-specific block, using a patch size of $p = 1$, the default window sizes of $(2, 6, 12)$, and varying embed_dim and num_heads as reported in Table 5.

**MeshGraphNet**    We formulate a periodically connected graph in east-west direction to apply MeshGraphNet and follow Fortunato et al. (2022) by encoding the distance and angle to neighbors in the edges. We employ four processor and two node/edge encoding and decoding layers and set hidden_dim = 32 for processor, node encoder, and edge encoder, unless overridden (see Table 5).

**GraphCast**    The original GraphCast model operates on a $0.25\,°$ resolution and implements six hierarchical icosahedral layers. As we run on a much coarser $5.625\,°$ resolution, we can only employ a three-layered hierarchy and employ three- and four-dimensional mesh and edge input nodes in four processor layers while varying the hidden channel size of all internal nodes according to the values reported in Table 5. Taking NVIDIA's Modulus implementation of GraphCast in PyTorch,[14] we are constrained to use a batch size of $b = 1$. For a comparable training process, we tried gradient accumulation over 16 iterations (simulating $b = 16$ as used in all other experiments), but obtained much worse results compared to using $b = 1$. We train GraphCast models with $b = 1$ and report the better results.

**FNO**    With FNO2D and TFNO2D we compare two autoregressive FNO variants, which perform Fourier operations on the spatial dimensions of the input and iteratively unroll a prediction along time into the future. While fixing the lifting and projection channels at 256, we vary the number of Fourier modes, channel depth, and number of layers according to Table 5.

**FourCastNet**    To diminish patching artifacts, we choose a patch size of $p = 1$ (see Appendix B.3 for an ablation with larger patch sizes), fix num_blocks = 4, enable periodic padding in the horizontal

---

[14]https://github.com/NVIDIA/modulus/tree/main/modulus/models/graphcast.

Table 5: Model configurations for WeatherBench experiments partitioned by model and number of parameters (trainable weights). For configurations that are not specified here, the default settings from the respective model config files are applied, e.g., `ConvLSTM` employs the default from `configs/model/convlstm.yaml`, while overriding `hidden_sizes` by the content of the "Dim." column of this table. Details are also reported in the respective model paragraphs of Appendix B.1.

| Model | #params | Enc. | Dim. | Dec. | Model | Dim. (hidden sizes) |
|---|---|---|---|---|---|---|
| ConvLSTM Cyl/HPX | 50 k | 3 × Conv2D with tanh() | 4 × 13 | 1 × Conv2D | U-Net Cyl | [1, 2, 4, 8, 8] |
| | 500 k | | 4 × 40 | | | [3, 6, 12, 24, 48] |
| | 1 M | | 4 × 57 | | | [8, 16, 32, 64, 128] |
| | 2 M | | 4 × 81 | | | [12, 24, 48, 96, 192] |
| | 4 M | | 4 × 114 | | | [16, 32, 64, 128, 256] |
| | 8 M | | 4 × 162 | | | [23, 46, 92, 184, 368] |
| | 16 M | | 4 × 228 | | | [33, 66, 132, 264, 528] |
| | 32 M | | 4 × 323 | | | [65, 130, 260, 520, 1040] |
| | 64 M | | 4 × 457 | | | [90, 180, 360, 720, 1440] |
| | 128 M | | — | | | [128, 256, 512, 1024, 2014] |

| Model | #params | Dim. | Model | Dim. (hidden sizes) |
|---|---|---|---|---|
| (T)FNO2D L4 | 50 k | 8 | U-Net HPX | [5, 10, 20, 40] |
| | 500 k | 27 | | [16, 32, 64, 128] |
| | 1 M | 38 | | [23, 46, 92, 184] |
| | 2 M | 54 | | [33, 66, 132, 264] |
| | 4 M | 77 | | [46, 92, 184, 368] |
| | 8 M | 108 | | [65, 130, 260, 520] |
| | 16 M | 154 | | [92, 184, 368, 736] |
| | 32 M | 217 | | [130, 260, 520, 1040] |
| | 64 M | 307 | | [180, 360, 720, 1440] |
| | 128 M | 435 | | [256, 512, 1024, 2014] |

| Model | #params | Dim. | #layers | Model | Dim. |
|---|---|---|---|---|---|
| FourCastNet | 50 k | 52 | 1 | SFNO | 13 |
| | 500 k | 168 | 2 | | 34 |
| | 1 M | 168 | 4 | | 61 |
| | 2 M | 236 | 4 | | 86 |
| | 4 M | 252 | 6 | | 117 |
| | 8 M | 384 | 6 | | 171 |
| | 16 M | 472 | 8 | | 242 |
| | 32 M | 664 | 8 | | 343 |
| | 64 M | 940 | 8 | | 485 |
| | 128 M | 1332 | 8 | | 686 |

| Model | #params | #heads | Dim. | #blocks | #lrs/blck | Model | #heads in layers | Dim. |
|---|---|---|---|---|---|---|---|---|
| Swin-Transformer (Cyl/HPX) | 50 k | 4 | 12 | 2 | 2 | Pangu-Weather | — | — |
| | 500 k | 4 | 40 | 2 | 4 | | [2, 2, 2, 2] | 12 |
| | 1 M | 4 | 60 | 2 | 4 | | [2, 4, 4, 2] | 24 |
| | 2 M | 4 | 88 | 2 | 4 | | [4, 8, 8, 4] | 32 |
| | 4 M | 4 | 124 | 2 | 4 | | [6, 12, 12, 6] | 60 |
| | 8 M | 4 | 88 | 3 | 4 | | [6, 12, 12, 6] | 96 |
| | 16 M | 4 | 60 | 3 | 4 | | [6, 12, 12, 6] | 144 |
| | 32 M | 4 | 84 | 4 | 4 | | [6, 12, 12, 6] | 216 |
| | 64 M | — | — | — | — | | [6, 12, 12, 6] | 312 |
| | 128 M | — | — | — | — | | — | — |

| Model | #params | $D_{processor}$ | Model | $D_{processor}$ |
|---|---|---|---|---|
| Mesh-GraphNet | 50 k | 34 | GraphCast | 31 |
| | 500 k | 116 | | 99 |
| | 1 M | 164 | | 140 |
| | 2 M | 234 | | 199 |
| | 4 M | 331 | | 282 |
| | 8 M | 469 | | 399 |
| | 16 M | 665 | | 565 |
| | 32 M | — | | 799 |

spatial dimensions, and keep the remaining parameters at their default values while varying the number of layers and channels as specified in Table 5.

**SFNO** Our first attempts of training `SFNO` yielded disencouraging results and we found the following working parameter configuration. The internal grid is set to `equiangular`, the number of layers counts four, while `scale_factor`, `rank`, and `hard_thresholding_fraction` are all set to 1.0 (to prevent further internal downsampling of the already coarse data). We discard position encoding and do not use any layer normalization, eventually only varying the model's embedding dimension according to Table 5.

### B.2 PROJECTIONS ON CLIMATE SCALES

Here, we share investigations on how stable the different architectures operate on long-ranged rollouts up to 365 days and beyond.

**365 Days Rollout** In Figure 13, we visualize the geopotential height $Z_{500}$ states generated by different models after running in closed loop for 365 days. For each model family, one candidate is selected for visualization (among three trained models over all parameter counts), based on the smallest RMSE score in $\Phi_{500}$, averaged over the twelfth month into the forecast. `SwinTransformer`, `FourCastNet`, SFNO, `Pangu-Weather`, `MeshGraphNet`, and `GraphCast` produce qualitatively reasonable states. The predictions of `ConvLSTM`, `U-Net`, `FNO`, and `TFNO` contain severe artifacts, indicating that these models are not stable over long-time horizons and blow up during the autoregressive operation. This is also reflected in the geopotential height progression over one year (Figure 14), where unstable models deviate from the verification data with increasing lead time. Figure 14 also reveals undesired behavior of `MeshGraphNet`, seemingly imitating `Persistence`, which results in a reasonable state after 365 days, but represents a useless forecast that does neither exhibit atmospheric dynamics nor seasonal trends.

In accordance with the qualitative evaluation of zonal wind patterns in Figure 5, we provide a quantitative RMSE comparison of how different models predict Trade Wind, South Westerlies, and Global wind dynamics in Figure 15. Only `SFNO`, `GraphCast`, `FourCastNet`,

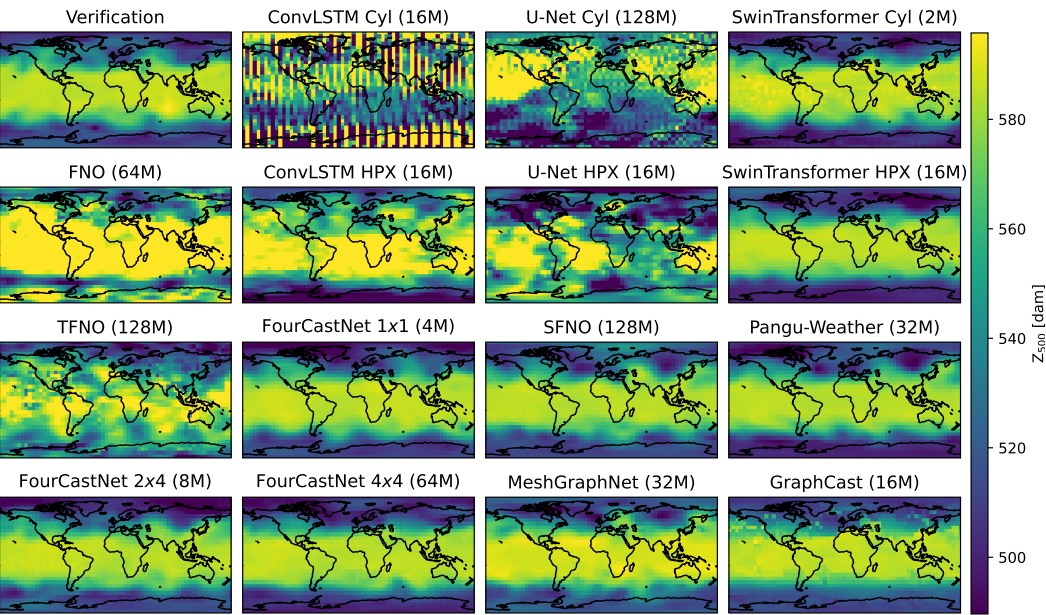

Figure 13: Snapshots of $Z_{500}$ predictions of different models at a lead time of 365 days, giving rise to a first differentiation between stable and unstable models.

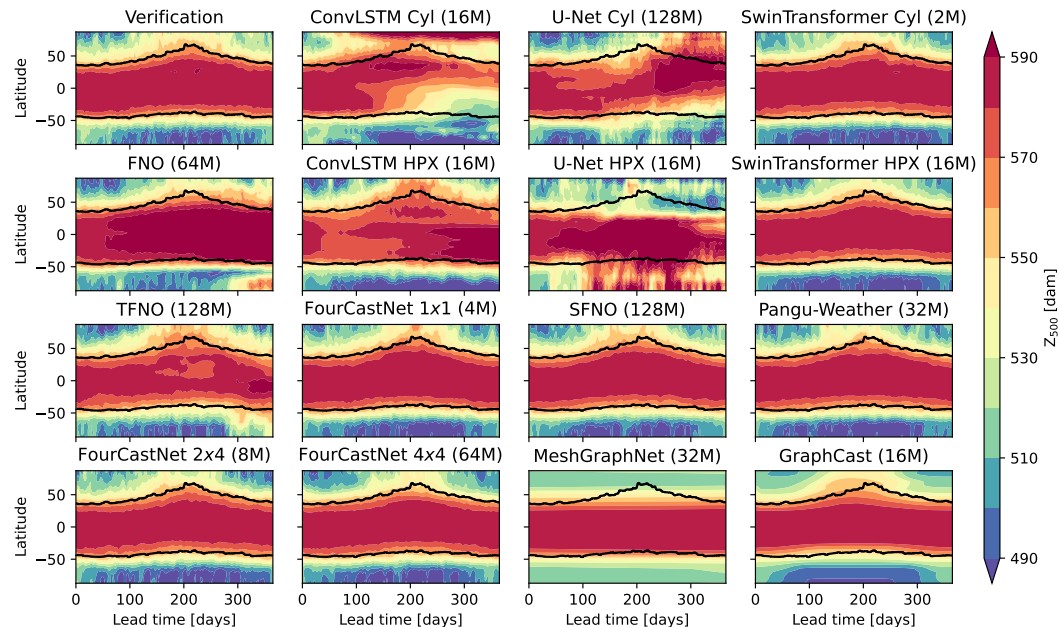

Figure 14: Zonally averaged $Z_{500}$ forecasts of different models initialized on Jan. 01, 2017, and run forward for 365 days. The verification panel (top left) illustrates the seasonal cycle, where lower air pressures are observed on the northern hemisphere in Jan., Feb., Nov., Dec., and higher pressures in Jul., Aug., Sep. (and vice versa on the southern hemisphere). The black line indicates the $540\,\mathrm{dam}$ (in decameters) progress and is added to each panel to showcase how each model's forecast captures the seasonal trend.

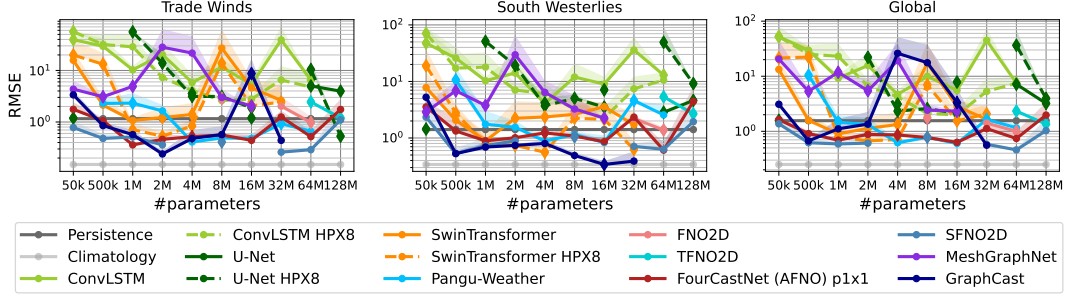

Figure 15: RMSE scores of different models predicting one-year averages of $U_{10}$ wind in three regions for various parameter configurations. From left to right: Trade Winds north and south of the equator, South Westerlies in the southern mid-latitudes, and an average over the entire globe. Errors are calculated after averaging predictions and verification over the entire year and the respective region. Diamond-shaped markers indicate that either one or two out of three trained models exceed a threshold of $100\,\mathrm{ms}^{-1}$ wind speed RMSE, and are then ignored in the average RMSE computation. Missing entries relate to situations, where none of the three trained models score below the threshold.

`Pangu-Weather`, and `SwinTransformer` outperform the `Persistance` baseline, yet without beating `Climatology`.

**50 Year Rollouts**   To investigate model drifts on climate time scales and further examine the stability of DLWP models, we run the best candidate per model family from the previous section for 73,000 autoregressive steps, resulting in forecasts out to 50 years. In Figure 16, we visualize longitude-latitude-averaged geopotential (left) and South Westerlies (right) predictions. Already

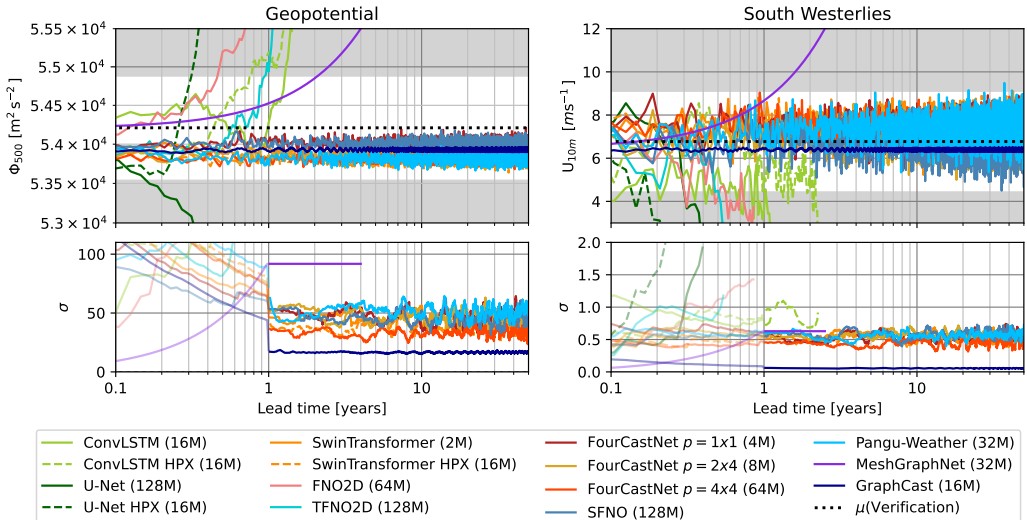

Figure 16: Top: Spatially averaged geopotential ($\Phi_{500}$, left) and South Westerlies ($U_{10m}$, right) predictions of selected candidates over 50 years. Shaded-areas depict intervals of $\pm 0.2$ (for $\Phi_{500}$) and $\pm 0.4$ (for $U_{10m}$) standard-deviations from the mean. Bottom: Annually averaged standard deviation progression over time of the statistics in the top panels. Lines are terminated once they exceed the y-limits in the top panels.

in the very first prediction steps (not visualized), all models drop to underestimate the average geopotential of the verification (black dotted line), which leads to large annually-averaged standard deviations in the first year. In line with previous findings, `SwinTransformer`, `FourCastNet`, `SFNO`, `Pangu-Weather`, and `GraphCast` prove their stability, now also on climate scale, without exhibiting model drifts (lines in the top panels of Figure 16 oscillate around a model-individual constant). Although suggested by the top panels, the models do not show an increase of standard deviation, as emphasized in the bottom panels, where $\sigma$ of the stable models also does not exhibit drifts.

**Power Spectra**   Another tool to evaluate the quality of weather forecasts is to inspect the frequency pattern along a line of constant latitude (Karlbauer et al., 2023; Nathaniel et al., 2024). In particular, the power analysis determines the frequencies that are being conserved or lost. A model that produces blurry predictions, for example, converges to climatology (regression to the mean) and looses high-frequencies. We contrast power spectra at five different lead times of one day, one week, one month, one year, and 50 years for the most promising models in Figure 17. Confirming the stability of `SwinTransformer`, `FourCastNet`, `SFNO`, `Pangu-Weather`, and `GraphCast` once again.

### B.3   In Depth Analysis of FourCastNet, SFNO, and FNO

**FourCastNet Ablations**   Surprised by the competitive results of `FourCastNet` and comparably poor performance of `SFNO`, we take a deeper look into these architectures to understand the difference in their performance. We would have expected `SFNO` to easily outperform its predecessor `FourCastNet`, since the former model implements a sophisticated spherical representation, naturally matching the source of the weather data. When replacing the core processing unit in `FourCastNet` with FNO and SFNO variants, we again observe best results for vanilla `FourCastNet` with its AFNO block as core unit, as reported in the top row of Figure 18.

In subsequent analyses, we vary `FourCastNet`'s patch size and observe two main effects, reflecting the resolution available in `FourCastNet` and the aspect ratio that ideally should match the aspect ratio of the data. When employing a patch size of $p = 1 \times 2$, for example, we observe best results, even outscoring the finer resolved `FourCastNet` with $p = 1 \times 1$. Respective results are provided in the bottom row of Figure 18.

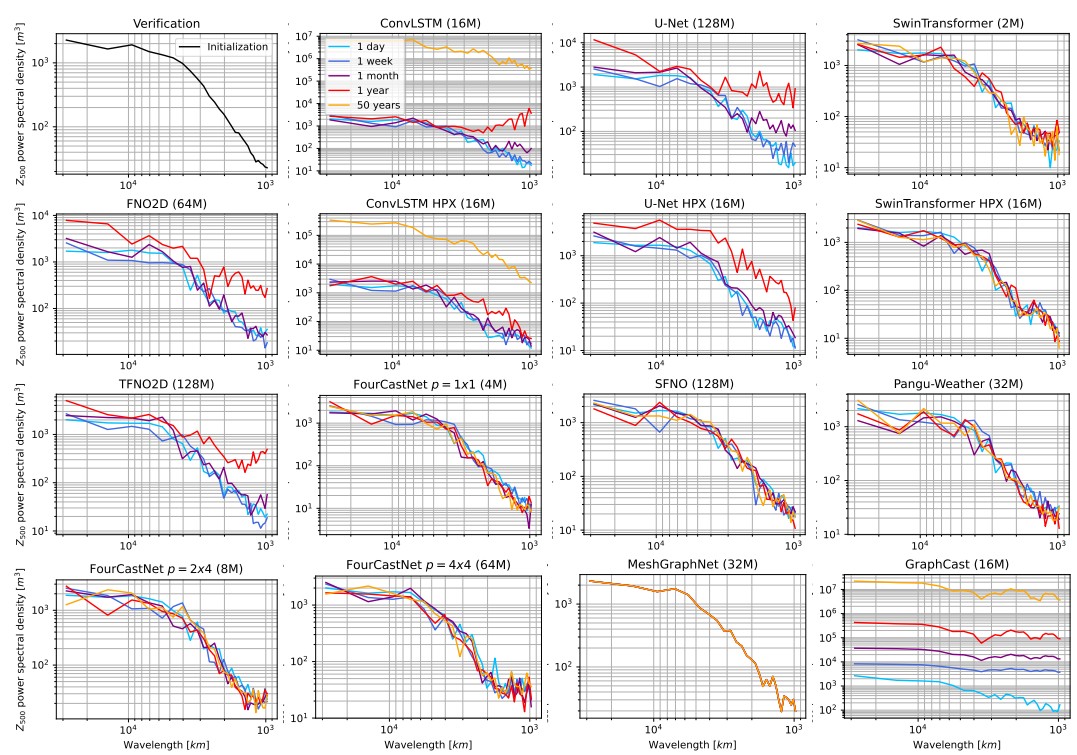

Figure 17: Power spectra of selected models at different lead times of one day (light blue), one week (dark blue), one month (purple), one year (red), and 50 years (orange). The top left panel shows the spectrum at initialization time; other panels represent a DLWP model each. Note the individual y-axis limits per panel.

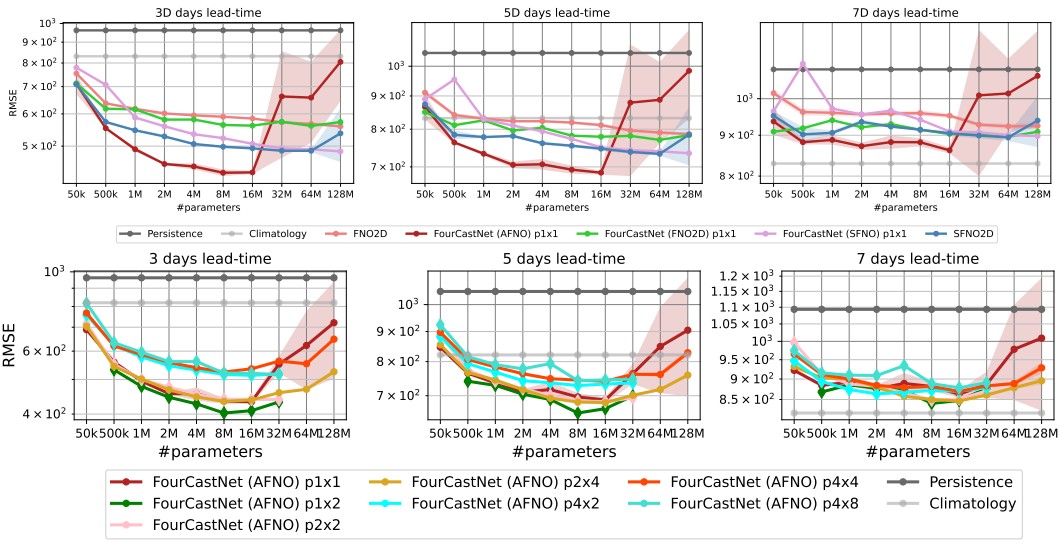

Figure 18: RMSE scores of different FourCastNet formulations on $Z_{500}$ vs. the number of parameters. Panels in the top row show results for FourCastNet when replacing the core AFNO forecasting-block with alternatives such as FNO and SFNO. The bottom row showcases the model error resulting for different patch sizes employed in the standard FourCastNet implementation. Triangle markers indicate statistics that were computed from less then three model seeds.

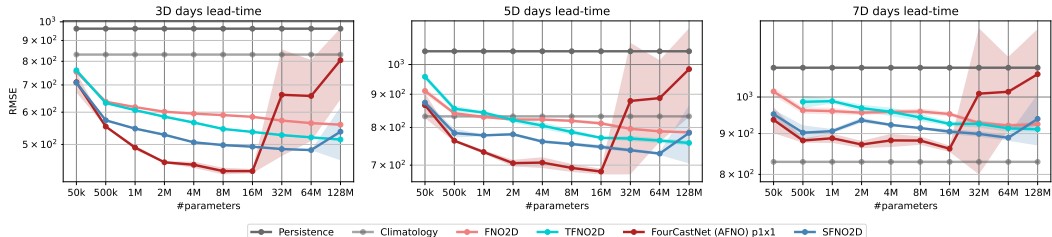

Figure 19: RMSE scores of selected FNO-based models on $Z_{500}$ vs. the number of parameters. While TFNO2D performs slightly better than FNO2D, these two architectures are outperformed by the more sophisticated FourCastNet and SFNO models.

**(T)FNO Comparison**   For a comparison of FNO and TFNO, we add another RMSE over parameters plot in Figure 19, which demonstrates the superiority of TFNO2D over FNO2D. We discard the 3D variants from the Navier-Stokes experiments due to their poorer performance compared to the 2D variants. To facilitate comparisons with FourCastNet and SFNO (both building on FNO), we also include their scores to the panels and observe the sophisticated models to be superior to the plain FNO architectures, which justifies the design choices in FourCastNet and SFNO for atmospheric state prediction, i.e., patching and spherical representation of the Earth.

### B.4   ADDITIONAL RESULTS

**Evaluations Beyond Geopotential and RMSE Metric**   To verify our results that were mostly obtained from statistics on the geopotential field, we provide additional RMSE-over-parameters plots in the second rows of Figure 20 (air temperature $2\,\mathrm{m}$ above ground), Figure 21 (zonal wind $10\,\mathrm{m}$ above ground), and Figure 22 (geopotential at $500\,\mathrm{hPa}$), analogously to Figure 2. We include a similar plot in the first rows of those plots that show the anomaly correlation coefficient (ACC)-over-parameters plot. Both the results on $T_{2m}$ and on the ACC metric support our findings, showing the superiority of ConvLSTM, FourCastNet, and SwinTransformer on short-to-mid-ranged forecasts. While the model ranking on the $T_{2m}$ variable follows the ranking on $\Phi_{500}$ in Figure 2, we particularly observe better results for SFNO, now being on par with other methods, especially on larger lead times.

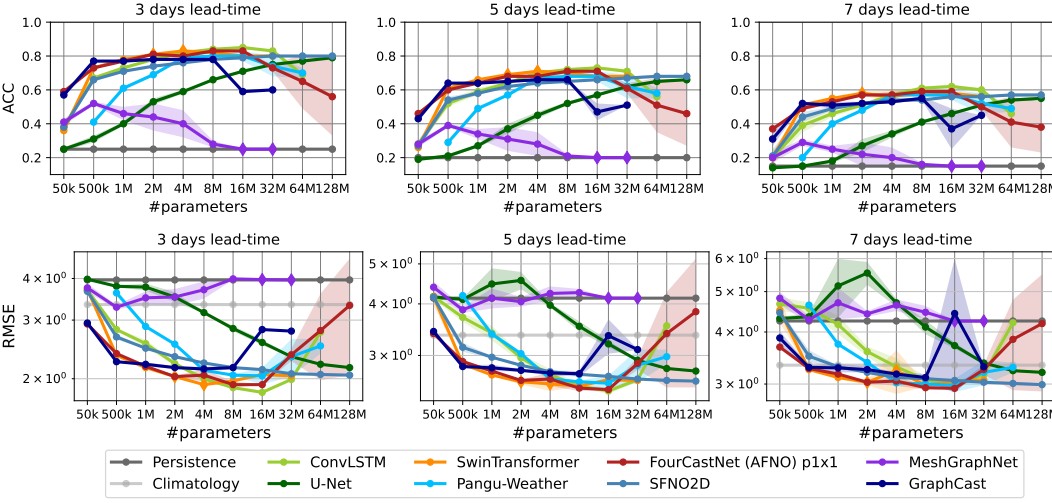

Figure 20: ACC (top) and RMSE (bottom) scores of all models on $T_{2m}$ vs. the numbers of parameters. Triangle markers indicate statistics that were computed from less then three model seeds.

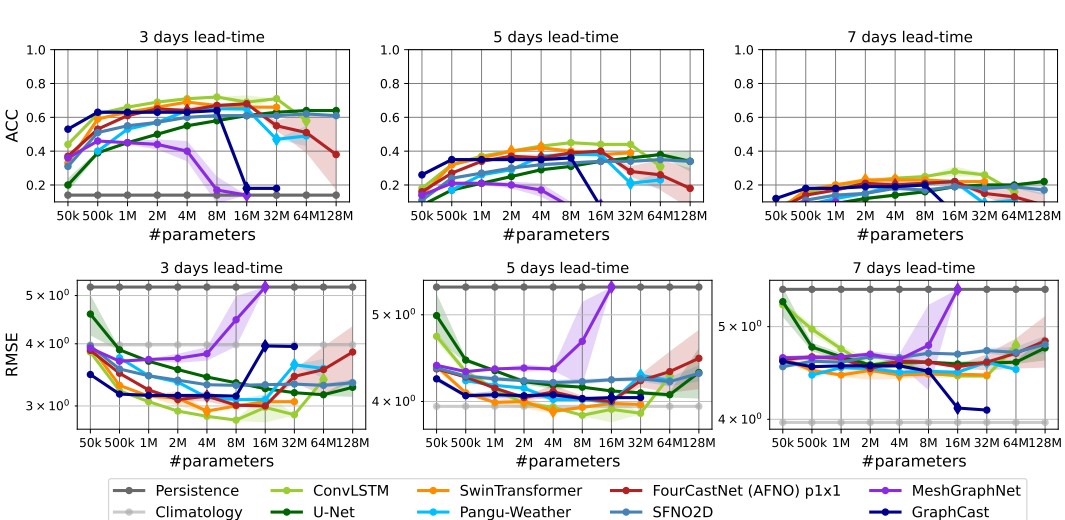

Figure 21: ACC (top) and RMSE (bottom) scores of all models on $U_{10m}$ vs. the numbers of parameters. Triangle markers indicate statistics that were computed from less then three model seeds.

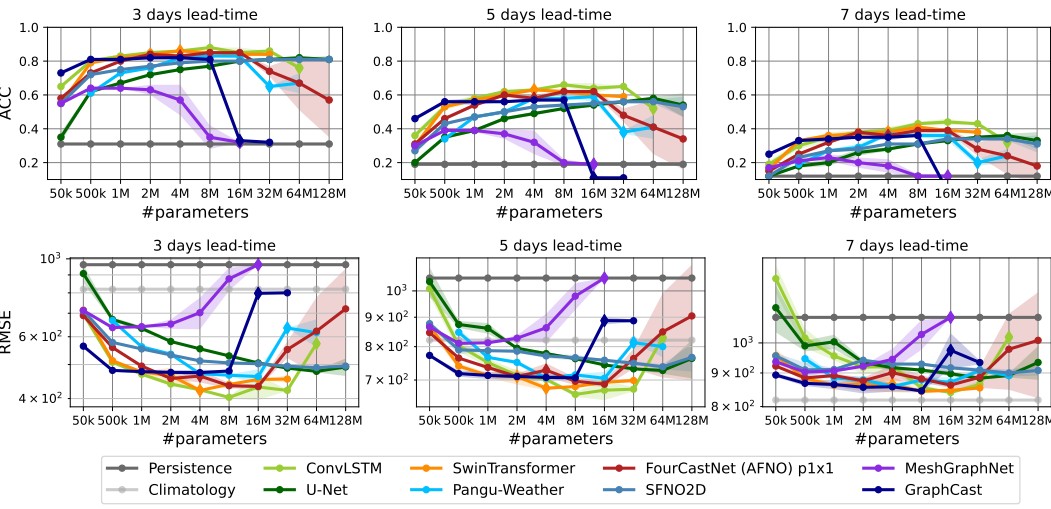

Figure 22: ACC (top) and RMSE (bottom) scores of all models on $Z_{500}$ vs. the numbers of parameters. Triangle markers indicate statistics that were computed from less then three model seeds.

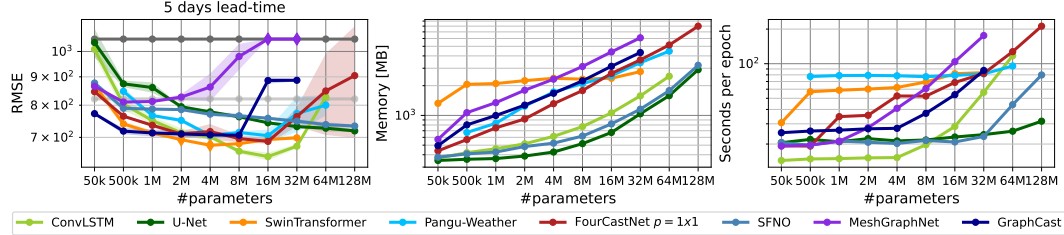

Figure 23: Memory consumption (center) and runtime (right), along with RMSE scores on $\Phi_{500}$ for the core models in our WeatherBench comparison. Log-scale on all axes.

**Runtime and Memory**    Similarly to our runtime and memory consumption analysis for the Navier-Stokes experiments (cf. Figure 8), we record the time in seconds for each model to train for one epoch with a batch size of $b = 1$. At the same time, we track the memory consumption in MB and report results, along with the five-day RMSE on $\Phi_{500}$ in Figure 23.

**ConvLSTM Training Progress**    To understand whether ConvLSTM models overfit in the high parameter count (as suggested in Figure 2), we inspect and visualize the training and validation curves of a $16\,\mathrm{M}$ and a $64\,\mathrm{M}$ parameter model in Figure 24. Seeing that both the validation and the training curves of the ConvLSTM $64\,\mathrm{M}$ parameter model show a similarly stalling behavior, we conclude that these models do not overfit, and instead fail to find a reasonable optimization minimum during training.

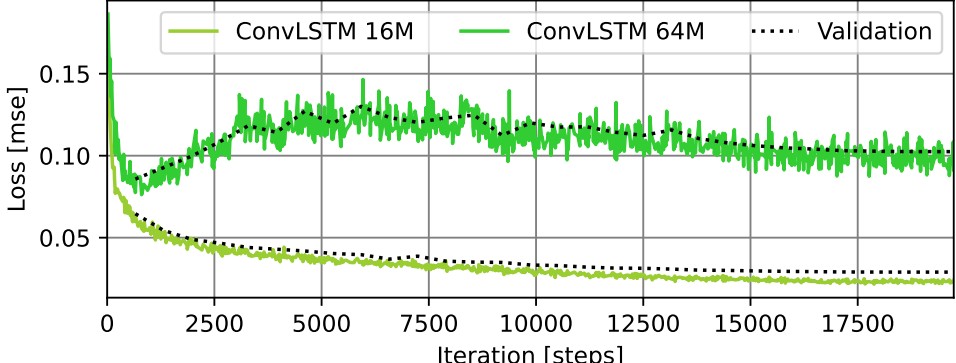

Figure 24: Training and validation error convergence curves of ConvLSTM with 16 and 64 million parameters.

