# OpenReview forum: "Comparing and Contrasting Deep Learning Weather Prediction Backbones on Navier-Stokes and Atmospheric Dynamics"
_ICLR.cc/2025/Conference — Submitted to ICLR 2025_

### Official Review · Reviewer_3TqW · 2024-10-22

**Soundness:** 4
**Presentation:** 4
**Contribution:** 3
**Rating:** 8
**Confidence:** 4

**Summary:**

This paper provides a fair comparison of the performance of widely used deep learning models for weather prediction. The authors standardize parametric settings, inputs, outputs, and training methods across models, and evaluate their performance using Navier-Stokes dynamics simulations, as well as medium- and long-term weather prediction. The study highlights each model's strengths and weaknesses.

**Strengths:**

This paper addresses a significant gap in the field, as there is currently no comprehensive and fair comparison of DLWP models. While many studies claim superior performance for their models, it remains unclear whether this is due to the backbone architecture, diagnostic variables, training, or inference strategies. By focusing specifically on the backbone models, the authors conduct rigorous experiments to empirically assess their forecasting potential. The findings are valuable for the DLWP field, offering novel insights, such as the superior performance of ConvLSTM, differences between Pangu-Weather and SwinTransformer, and the influence of FourCastNet's patch size.

**Weaknesses:**

1. The experiment of synthetic Navier-Stokes simulations seem to play a limited role. As noted in line 160, there is a significant gap between the univariate Navier-Stokes simulation and real atmospheric dynamics. Given that the authors aim to evaluate backbone models' performance in the more complex weather forecasting task in section 3.2, dedicating one-third of the main text to the simpler univariate Navier-Stokes simulation seems unnecessary. In light of the results from section 3.2, the findings from section 3.1 appear less relevant and, in fact, somewhat confusing:

    (a)  Section 3.1 highlights the superiority of TFNO, yet this model is absent from section 3.2. Since TFNO appears in Figures 13, 14, and 15, its performance in RMSE metric should have been assessed by the authors.

    (b) In Figures 13, 14, and 15, TFNO, ConvLSTM, and UNet underperform compared to other models.

    (c) In Figure 16, models that perform well in section 3.1 (TFNO, ConvLSTM, UNet) exhibit poor stability.

2. In section 3.2, the authors evaluate the performance of different backbone models using the 5.625 deg ERA5 dataset. The experiments provide limited guidance for selecting backbone models for operational weather prediction, which typically relies on the 0.25 deg ERA5 dataset. However, large-scale experiments at this resolution are obviously costly, so this is an existential but understandable drawback :) .

**Questions:**

Overall, as the first paper to provide a fair comparison of various DLWP models, this work has the potential to make a significant contribution to the field. However, I recommend the authors reconsider the emphasis placed on section 3.1 and expand the experimental results in section 3.2, particularly by including RMSE metrics for variables such as t850, u10, and v10.

Here are other questions:
1. Lines 291-293: The ACC metric is not provided in section 3.2.1. Additionally, section 3.2.1 presents RMSE metrics for geopotential only up to 7 days, not 14. Given the presence of 8 prognostic variables, it would be beneficial to include the RMSE and ACC metrics for all variables, potentially in the appendix in a format similar to Figure 2.
2. In Figures 2 and 18, the authors observe that SwinTransformer outperforms Pangu-Weather in terms of RMSE. To my knowledge, the primary difference between Pangu-Weather and SwinTransformer is the use of Earth-specific positional bias in Pangu-Weather. Intuitively, this difference alone should not lead to such a performance gap. I suggest that the authors standardize other hyperparameters (e.g., layers, embedding dimensions) between Pangu-Weather and SwinTransformer, and present additional results, such as RMSE for geopotential with 1M parameters, to clarify this discrepancy.
3. Lines 339-340: The authors limit the optimization cycle to 24 hours (4 steps). While there is no established standard for optimization lead time, I question whether ConvLSTM, being the only RNN-based model, is particularly sensitive to this hyperparameter. State-of-the-art models like FourCastNet (2 steps), Pangu-Weather (1 step), and GraphCast (1 step in pretraining) use shorter optimization cycles. Training with 4 steps may become resource-intensive at higher spatial resolutions, which could be a limitation of ConvLSTM.
4. Lines 340-341: The authors evaluate the backbone models using initial conditions at 00z. I wonder if fixing the initial time at 00z simplifies the overall weather prediction task. Could the authors test whether models trained on 00z initial conditions also perform well with 12z initial conditions in the test set?
5. Lines 489-490: In Figures 5 and 15, the authors note that SFNO performs well in predicting wind fields, accurately capturing real-world wind patterns. They attribute this to SFNO's adherence to physical principles. However, given SFNO's performance in Figure 2, I question whether this claim holds true for all prognostic variables.
6. Line 863: Since $x,y \in \mathbb{N}$, it follows that $x+y \in \mathbb{N}$. Therefore, in the authors’ setting, $f \equiv 0.1$. I think there must be some mistake. Otherwise, the Navier-Stokes simulation is too simple.
7. Line 1228-1229: The ‘]’ of heads in layers in Pangu-Weather is missing.
8. In Figure 14, why are the only two graph neural networks smoothed in Zonally averaged forecasts?
9. In Figures 2, 17, 18, and 19, I observe that the confidence intervals for some models, particularly FourCastNet, are notably wide across the three random seeds. Upon reviewing the code, I suspect this may be due to the gradient clipping, which is set equal to the learning rate ($\leq 10^{-3}$). When multiplied by the learning rate, the step size of the gradient descent ($||\eta *\text{Clip}(\nabla f)||_{2}$) is less than $1\times 10^{-6}$, which is likely too small for effective exploration of the parameter landscape. As a result, model performance may be highly dependent on initial parameters or random seeds. My question is, why was the gradient clipping value set equal to the learning rate? Is there a specific reference for this choice?
10. Line 684-686: unify the reference.

---

> ### Author Response · Authors · 2024-11-28
>
> Thank you very much for providing this extremely detailed and constructive review and for assessing our work so positively. We are glad that you recognize the significance of our controlled comparison study for the DLWP research community. In the following, we will respond to the weaknesses and questions you posed.
>
> ### Weaknesses
> **W1 Much Space for Navier-Stokes** Even though we agree that the results on the synthetic data play a subordinate role, we like the similarity of trends when comparing Figure 1 and Figure 2, which underlines that the two datasets share certain principles. We next touch on your (a) through (c). (a) We apologize that TFNO appeared discarded and have added an explicit comparison of FNO and TFNO on WeatherBench in Figure 19 of the updated manuscript, matching the results from Navier-Stokes, where TFNO > FNO. (b) and (c) That’s right: TFNO, ConvLSTM, and U-Net perform poorly in the _long rollout_ experiments on WeatherBench. We do not have a comparable long rollout experiment on the synthetic Navier-Stokes data.
>
> **W2 WeatherBench Resolution** Thanks for expressing your understanding of our compute considerations when deciding to use data at 5.625 degrees resolution. Please refer to our answer to Q1 of Reviewer BnFD for our argument on the choice of coarse resolution.
>
> ### Questions
> **Q1 RMSE Plots for T850, U10, V10** We understand that a full report of all prognostic variables is of interest for the community and are about to complete according evaluations. In the meantime, please refer to Figure 21, which features RMSE over parameters for U10. We are extending this analysis to T850, V10, Z250, Z700, and Z1000.
>
> **Q2 SwinTransformer vs Pangu-Weather** In fact, a more detailed analysis of the differences of SwinTransformer and Pangu-Weather would be insightful. However, we decided not to include additional ablation experiments in this study, as we have varied the blocks, heads, layers and dimensions of both architectures already to a reasonable extent, as spelled out in Table 5.
>
> **Q3 ConvLSTM** Arguably, the larger proportion of the GPU memory is occupied by data and not by the models, in particular when going to finer resolutions. However, moving towards 24h optimization is crucial to capture the circadian cycle and to train the model in handling its own output, which is key for long rollouts.
>
> **Q4 00z vs 12z Initialization** This is a very sharp observation and we had the same concern in earlier projects. We could alleviate these concerns back then by indeed initializing the models at noon and not finding substantial differences. It seemed like the 24h optimization cycle trained the model to handle arbitrary initialization times.
>
> **Q5 SFNO at U10** Inspecting U10 predictions out to seven days (Figure 21 in revised manuscript) in fact yields a different picture, where SFNO only performs mediocre anymore. In Figures 5 and 15, though, we are reflecting on long rollouts, where SFNO proves superior. This underlines SFNO’s strength in long and stable rollouts, while not mastering short- to mid-ranged lead times.
>
> **Q6 Navier-Stokes Simple** We double checked the forcing factor $f=0.1(sin(...$ as being the one we have used in our experiments. In experiment 1, the dynamics could be learned well by all models (albeit only with sufficient parameters), whereas the more turbulent setting in experiment 2 and 3 were more challenging. Thus, we do not think the dynamics were too simple.
>
> **Q7 Typo** Nice catch, we have added the closing parenthesis. Thanks for pointing it out.
>
> **Q8 Zonally Smoothed GNNs** MeshGraphNet tends to mimic persistence instead of predicting daily dynamics (mentioned around 1265–1267 in the revised manuscript) and GraphCast seems to follow a similar trend, albeit not as pronounced as in MeshGraphNet. This is now also visible in the power spectra of Figure 17.
>
> **Q9 Gradient Clipping** Gradient clipping played a capital role when optimizing larger models (on both datasets). Since we were using a cosine learning rate scheduler, we wanted the clipping to follow the magnitude of the gradient signal (we cannot point to a resource for this decision). Empirically, with a constant clipping rate, we observed more instabilities, likely due to relatively large gradients in late stages of the training. Thus, binding the gradient clipping to the learning rate resulted in better convergences and actually less blow ups in our setting.
>
> **Q10 Unify Reference** Looking at Saad et al, we could not find irregularities. Can you point us towards what you mean specifically?
>
> Please let us know if we missed to address a particular aspect of your review and questions. We are looking forward to further inspiring discussions.

---

> > ### Comment · Reviewer_3TqW · 2024-11-28
> >
> > Thanks to the author for the reply. Here are my additions to question 6 as well as question 10:
> >
> > **Q6** At the bottom of page 16 of the updated manuscript, there is '$f = 0.1 (\sin(2\pi(x+y)) + \cos(2\pi (x+y)))$, with $x, y \in [0, 1, ..., 63]$'. Does this indicate that $f \equiv 0.1$? If $f \equiv 0.1$, why did authors formulate $f$ in such a form?
> >
> > **Q10** Authors can refer to lines 671-673 (Pfaff et al) and lines 690-692 (Saad et al), the citation formats of ICLR are different.

---

> > > ### Author Response · Authors · 2024-11-29
> > >
> > > We highly appreciate your commitment in discussing our work. Thanks for clarifying on Q6 and Q10, we well understood your points now and respond below.
> > >
> > > **Q6** The choice of the forcing factor $f=0.1$ stems from our incentive to be comparable with the [Fourier Neural Operator paper](https://arxiv.org/abs/2010.08895) [1] for two reasons: We first wanted to verify whether we obtain similar results with our experimental setup as reported in [1], to afterwards exceed the comparison provided in [1] to other DLWP-related architectures, i.e., Transformers, GNNs, and ConvLSTM. In Section 5.3 of [1], the forcing function is introduced as $f\in L^2_{per}((0, 1)^2; \mathbb{R})$. similarly to another work on [Multiwavelet Operator Learning paper](https://proceedings.neurips.cc/paper/2021/hash/c9e5c2b59d98488fe1070e744041ea0e-Abstract.html) [2] we overall follow the parameter choice of [1] in our Navier-Stokes data generation, as emphasized in lines 187--188 of our manuscript.
> > >
> > > **Q10** We corrected the Saad et al. reference by removing _The Eleventh_ from the conference name. The reference now reads `Nadim Saad, Gaurav Gupta, Shima Alizadeh, and Danielle C Maddix. Guiding continuous operator
> > > learning through physics-based boundary constraints. In International Conference on Learning
> > > Representations, 2023`, which now is consistent with the other ICLR works we are citing. Thanks for pointing this out.
> > >
> > > Does our response resolve your questions?

---

> > > > ### Comment · Reviewer_3TqW · 2024-11-29
> > > >
> > > > Thanks to the authors for their responses. The author's replies are instructive and informative to me as well as researchers in the DLWP field (although I have not verified them, I tend to believe so). There are still some settings that confuse me (e.g., the threshold for Gradient Clipping is the same as the learning rate), but from the intention of this research, there is no reason for the authors to deliberately craft some settings or parameters.
> > > >
> > > > I have carefully read the review comments from other reviewers. As someone who has done similar experiments (training from scratch, harmonizing settings as well as parameters, etc.), I recognize the authors' conclusions (on intersecting, we are in general agreement) and can understand where the authors' contribution lies. Taking all these factors into account, I decided to maintain my given score and increase my confidence.

---

### Official Review · Reviewer_ijdW · 2024-11-02

**Soundness:** 2
**Presentation:** 2
**Contribution:** 2
**Rating:** 3
**Confidence:** 5

**Summary:**

The paper analyzes the performance of different network structures for weather prediction. Experiments were conducted on both synthetic and real data, and a benchmark was established. They also suggested network structures suitable for mid-term and long-term forecasting.

**Strengths:**

The design of the backbone network greatly affects the performance of machine learning models. This article provides an analysis of the backbone network's performance in weather forecasting.

**Weaknesses:**

This article seems like an experimental report. It includes introductions to several classic backbone networks, settings for two experimental datasets, and descriptions of the results. However, this paper lacks insights that previous work did not reveal.

**Questions:**

1. The authors introduced a new benchmark for weather forecasting, but they didn't clearly explain how it differs from previous research, such as in data construction and task definition.
2. The authors analyzed several backbone networks, but only showed some quantitative results without providing more insights, such as proposing new designs for backbone networks.
3. The authors used synthetic and real data to train these models, but they did not discuss the differences between these data and the data used by existing state-of-the-art models.
4. The number of model parameters used by the authors seems small, but current weather prediction models use a large number of parameters. With such a big difference in parameter count, is the conclusion reliable?
5. With only 1K and 10K samples in experiments 1 and 2, are these numbers too small? Can the conclusions be trusted?

---

> ### Author Response · Authors · 2024-11-28
>
> Thank you for taking the time to assess our manuscript. We appreciate your review, but want to clarify various aspects of your conclusions and questions.
>
> ### Limited Novelty
> In line with Reviewer 3TqW, stating that ``The findings are valuable for the DLWP field, offering novel insights…``, we strongly disagree with your concern that ``... this paper lacks insights that previous work did not reveal.`` We sketch a non-exhaustive list of new findings to the DLWP research community in our work:
> 1. Our study **uniquely offers a fair ``apples-to-apples'' comparison of DLWP models and their backbones _across parameter counts_ for the first time**, allowing a genuine assessment of the suitability of different models for different tasks in the context of atmospheric state prediction.
> 2. Importantly, we provide first estimates on **scaling behavior of DLWP architectures**, which not only apply to weather prediction but are of value for the larger deep learning community, as we consistently benchmark a large number of different architectures under rigorously controlled conditions.
> 3. In stark contrast to previous studies, **we find SFNO performing at average on short-to mid-ranged lead times out to 14 days**, while other architectures deliver more accurate results. We have been in exchange with the SFNO authors and acknowledge their time in helping us to spend more time in tuning SFNO (applying tweaks, e.g., using larger learning rates, removing positional embeddings, and using different latent meshes for the SFNO projections) compared to other architectures.
> 4. We find a **stable behavior for GraphCast, Pangu-Weather, and FourCastNet**, which all were disqualified as unstable for long-ranged predictions in their sophisticated formulation either in their own publication or in follow up analyses (Bonev et al., 2023, Karlbauer et al., 2024). It is crucial to understand that these methods actually can generate stable forecasts under certain choices of prognostic variables and hyperparameter. For example, our FourCastNet ablations in Appendix B.3 suggest a patch size that matches with the aspect ratio of the data, that is, 1:2 for lat-lon. **Our finding features patch sizes of $p=1\times2$ more expressive over $p=1\times1$**, despite the reduced availability of information (see Figure 18, bottom).
> 5. Our **exhaustive tests on long-ranged rollouts** (Section 3.2.2) and reproduction of physical properties (Section 3.2.3) for the first time provide estimates on the stability of _various_ DLWP models (beyond SFNO).
> 6. In line with findings from NLP, we observe an easy-to-optimize behavior of transformers on weather dynamics, whereas other architectures (particularly GNNs) require more fine tuning.
>
> Kindly point us to other work that has made these contributions before us.
>
> ### Questions
> **Q1 Benchmark Description**
> We do provide detailed information about data selection and construction in paragraph **Data Selection** in Section 3.2, clearly spelling out that our benchmark ressembles a subset of WeatherBench. Also, we are precise in the research questions (i.e., task description) of our benchmark by enumerating three concrete goals at the beginning of Section 3.2. Moreover, in lines 59–63 of our manuscript, we clearly differentiate our benchmark from previous work.
>
> **Q2 New Design Proposals for DLWP**
> The primary goal of our study is to evaluate existing DLWP architectures and their backbones, not to propose new design choices. We do agree that new design choices are of interest to the DLWP community, yet this goes beyond the scope of this work. Both in our Introduction (lines 64–69) and Discussion (second to last paragraph), we spell out what architecture type has the largest potential for particular downstream tasks and encourage DLWP practitioners to adhere to respective models when aiming to work on certain tasks, e.g., using ConvLSTM for forecasts out to seven days.
>
> **Q3 Synthetic and Real-World Data Discussion**
> We do provide a thorough motivation of using Navier-Stokes and WeatherBench at the beginning of Section 3.1 and Section 3.2, respectively, which also explains the differences between these datasets. Furthermore, we detail what data respective DLWP models use in the very first paragraph of the Introduction.
>
> **Q4 Small Parameter Counts**
> State-of-the-art DLWP models like Pangu-Weather, GraphCast, and U-Net consist of 64M, 21M, and 10M parameters (note that Pangu-Weather reports 256M parameters in total when training four separate models for different lead times). Thus, the number of parameters in our experiments, ranging from 50k to 128M, very well aligns with that of SOTA DLWP models.
>
> **Q5 1k and 10k Samples too Few**
> We address this question precisely in Figure 12, showing that TFNO3D with large parameter counts started to overfit on the Navier-Stokes dataset with 1k samples. The right-hand panels of Figure 12 proof that increasing the number of sequences to 10k resolved the overfitting issue.

---

> > ### Comment · Reviewer_ijdW · 2024-12-02
> >
> > Thank you for the author's response, but most of my concerns remain unresolved.
> >
> > I strongly agree that conducting a comprehensive evaluation of DLWP in a fair setting is meaningful. However, the contribution of this paper is very unclear, as I mentioned in my previous comments.
> > 1. This paper appears to be a poorly organized experimental report. The paper's main content merely demonstrates the performance variations of different backbones on DLWP tasks, with no design in the methods section, making it difficult to grasp the core contribution of this paper.
> > 2. The authors continually argue that they are the first to propose a fair comparison of DLWP models, estimate the scaling behavior of DLWP architectures, and assess the stability of various DLWP models, finding that SFNO performs at an average level on short-to mid-range lead times up to 14 days. However, a high-quality paper often has 1-2 high-quality core contributions that are sufficient to recommend acceptance. This article makes it hard to identify the most important contribution.
> > 3. The authors claim to provide the first estimates of the scaling behavior of DLWP architectures, yet the specific contributions remain unclear. They only offer some unconvincing experimental observations, which do not provide insight.
> > 4. The authors claim to have found stable behavior for GraphCast, Pangu-Weather, and FourCastNet in long-term forecasting. However, does the setting used for this conclusion align with these methods? For example, the number of model parameters, the numbers and resolution of samples in the dataset, etc. These basic setting differences make it hard to be convinced by the author's conclusions.
> > 5. Is the purpose of this paper to create a new benchmark (e.g., WeatherBench) to allow researchers to design backbones in a more fair setting? Yet the authors seem to have only used a subset of WeatherBench and did not involve the benchmark design.
> > 6. Is the author proposing a new systematic evaluation framework involving new metrics or evaluation mechanisms? It seems difficult to find any new contributions in the evaluation.
> > 7. Is it through extensive experimental analysis to propose a new, effective, and powerful backbone? From the author's response, it appears they have no such intention. The author claims this part is beyond the scope of the study, yet it seems hard to find contributions in other points.
> > 8. DLWP often involves large-scale model parameters, yet the authors only compared small model parameters, making their conclusions hard to believe and I fear it may mislead the community. I also know that systematic evaluation of large-scale models may bring greater computational costs, but the authors should have some experiments to support it rather than avoiding this commonly used model scale in DLWP. For example, the results shown in Table 1, with model parameter scales of 5k, 50k, 500k, hardly generate practical significance.
> > 9. The issue of dataset scale remains a significant weakness, and I find it hard to understand conducting experimental analysis with such a small amount of data in DLWP. For example, increasing the dataset from 1K to 10K to solve overfitting issues is common sense and does not provide any insight.

---

> > > ### Author Response · Authors · 2024-12-03
> > >
> > > Thank you for getting back to us. When reading through your enumeration, we identify two core aspects, which we have addressed in our first answer already. We re-emphasize our position in the following and specify in parentheses to which of your comments our answer relates.
> > >
> > > ### Unclear Contribution (C1, C2, C3, C5, C6, C7)
> > > The contributions of our work are clarified in our manuscript at the end of the Introduction, which is common practice in ICLR papers, and we also provide the following motivation of our work (lines 64--69 of our revised manuscript):
> > >
> > > `With our analysis, we also seek to motivate architectures that have the greatest potential in addressing
> > > downsides of current DLWP models. To this end, we focus on three aspects: (1) short- to mid-ranged
> > > forecasts out to 14 days; (2) stability of long rollouts for climate lengthscales; and (3) physically
> > > meaningful prediction.`
> > >
> > > We have spelled out a list of contributions of our work in our previous answer and we kindly ask you to relate to that post.
> > >
> > > ### Too Few Parameters and Data (C4, C8, C9)
> > > We do not agree that the parameter count in our benchmark is not representative. Please relate to our answer in **Q4 Small Parameter Counts** of our first answer, which we repeat in the following:
> > > _State-of-the-art DLWP models like Pangu-Weather, GraphCast, and U-Net consist of 64M, 21M, and 10M parameters (note that Pangu-Weather reports 256M parameters in total when training four separate models for different lead times). Thus, the number of parameters in our experiments, ranging from 50k to 128M, very well aligns with that of SOTA DLWP models._
> > >
> > > The parameter ranges you are reporting from Table 1 relate to our first batch of experiments, which relates to synthetic Navier-Stokes dynamics and not to Deep Learning Weather Prediction. The same applies to the dataset size of 1k to 10k, which we believe have addressed accurately in **Q5 1k and 10k Samples too Few** of our first answer.
> > >
> > > ### How Can We Improve?
> > > We would appreciate any concrete suggestion of how we can improve our work. Instead, with statements such as `a poorly organized experimental report` or `unconvincing experimental observations`, we have a hard time in improving our report. Please detail for what reasons you think our report is organized poorly and why experimental observations are unconvincing.
> > >
> > > Thank you for your patience and feedback.

---

### Official Review · Reviewer_F4LR · 2024-11-03

**Soundness:** 3
**Presentation:** 3
**Contribution:** 2
**Rating:** 5
**Confidence:** 4

**Summary:**

The paper provides a comparative study of various architectures such as U-Net, Transformers, Graph neural networks, ConvLSTM, and Neural Operators that have shown their potential to serve as backbones in Deep Learning Weather Prediction (DLWP) models. This work includes a systematic and detailed empirical analysis under controlled conditions, controlling for parameter count, training protocol, and prognostic variables. All the models are evaluated by benchmarking on two systems: synthetic Navier-Stokes and real-world weather datasets. The paper focuses on short-to-mid-ranged forecasts, long-ranged (climate length) forecasts, and physics-backed forecasts, intending to provide better architectural design choices supporting the DLWP research for various forecasting tasks. Based on their observation, ConvLSTM is better at short and mid-range forecasts on weather data. For stable long-ranged forecasts, aligned with physics principles, spherical representations such as in GraphCast and Spherical FNO, show superior performance.

**Strengths:**

- The experiments in this study are extensive, and the analysis is presented in a clear, organized manner. The details of the experiments are thoroughly explained and the set of models chosen for the comparison is justified well.
- The paper includes long-range forecasts for lead times as long as 365 days (and more), which is important and not included in most DLWP studies. These results can be insightful to this line of research.

**Weaknesses:**

- The spatial resolution used in the paper for global weather prediction is too coarse (5.625 degrees) as compared to the 0.25-degree resolution used in recent weather forecasting models such as FourCastNet, PanguWeather, and GraphCast.
- Using the backbones of the DLWP models for performing prediction on the Navier-stokes system does not seem very relevant to the contributions of this work. The paper also says “A direct transfer of the results from Navier-Stokes to weather dynamics is limited”. Moreover, FNO working so well on Navier-Stokes has already been shown before.
- The paper claims to be studying physically meaningful forecasts. This is a crucial aspect of weather forecasting and should be a critical factor in comparing models. However, the paper doesn’t go into much detail on this aspect. For instance, physics-based metrics and power spectrum plots [1] are needed to investigate if the models can capture small-scale (high-freqeuncy) features in their forecasts.
[1] Nathaniel, Juan, et al. "Chaosbench: A multi-channel, physics-based benchmark for subseasonal-to-seasonal climate prediction." 2024.

**Questions:**

- What is the justification behind using a coarse spatial resolution for weather prediction?
- The authors should add more on why they chose to evaluate and compare the models on the Navier-Stokes system.
- There needs to be more analysis to understand the physical soundness of various models. This should include physics-based plots/metrics as suggested before, and a discussion comparing models on this aspect of their forecasting skill.

---

> ### Author Response · Authors · 2024-11-28
>
> Thank you for taking the time to work through our manuscript and for providing such a detailed and constructive review. We are glad that you value our work as insightful for the community. Please find in the following our responses to your questions.
>
> **Q1 Spatial resolution**
> Using the 5.625 degrees resolution of WeatherBench has practical reasons. We expect all models to improve in performance, roughly by a constant factor, and thus decided to operate on the coarsest resolution to save compute. Earlier experiments (not associated with this research project) showed consistent improvement with finer resolution. In our manuscript, however, we are not aiming for producing state-of-the-art results, but to compare DLWP models under controlled conditions.
>
> **Q2 Navier-Stokes**
> Due to the higher complexity of real-world weather dynamics, we do not expect a direct transfer of results on synthetic Navier-Stokes data. As motivated at the beginning of Section 3.1, though, the Navier-Stokes dynamics do relate to atmospheric dynamics and thus constitute an appropriate dataset for an initial exploration. This is confirmed when comparing Figure 1 with Figure 2, outlining similar trends on the two datasets. Importantly, we emphasize in our Discussion that FNO works well on Navier-Stokes, but not as well when directly applied to WeatherBench (see lines 502–506 of our revised manuscript).
>
> **Q3 Power Spectra**
> We very much like your encouragement to perform spatial frequency analyses to investigate the physical soundness of model outputs. We thus computed power spectra for selected models (those, which turned out most promising in earlier analyses). The power spectra soundly match with our previous findings and are now contained in Figure 17 of the appendix of the revised manuscript.
>
> We are curious to hear back from you as to understand whether our responses clarified your questions and concerns.

---

> > ### Comment · Reviewer_F4LR · 2024-12-02
> >
> > I thank the authors for their response and for adding additional power spectra plots. I understand that comparing models for a higher-resolution prediction, such as 0.25 degrees, was not a goal of this work. However, it would be interesting and perhaps more useful to observe the ordering of the models in that case.
> >
> > Regarding the power spectra, I believe it is more meaningful to compare all the models in a single figure for a particular lead time (preferably with all energy values normalized to the first energy value to provide a consistent starting point for all models, e.g. in McCabe et al. 2023), rather than comparing across lead times for a single model. The current set of figures in Fig. 17 does not adequately 'compare and contrast the models,' which is more important than 'comparing the lead times' of a single model. Additionally, there should be more discussion on these plots, including an analysis of which model performs better at capturing high-frequency features in their forecasts. This discussion should preferably be included in the main paper (Section 3.2.3: Physical Soundness) rather than being limited to the appendix, where the authors only state: 'Confirming the stability of SwinTransformer, FourCastNet, SFNO, Pangu-Weather, and GraphCast once again.' I believe comparing the physical meaningfulness of model predictions is just as important as comparing them using error metrics such as RMSE, especially since the authors claim this as a contribution. Based on these considerations, I have decided to maintain my original score.

---

> > > ### Author Response · Authors · 2024-12-04
> > >
> > > We agree that a comparison on high-resolution data would be of great interest to the community as well, in particular when extending over our 5.625 degrees results and possibly confirming them.
> > >
> > > Encouraged by your concrete and constructive suggestion to overlay the spectrum plots of all models in one figure, we did so for different lead times. Please find our results in the anonymous repository that we provide in our manuscript. Concretely, the figures can be found at [this link](https://anonymous.4open.science/r/dlwp-benchmark-F88C/src/dlwpbench/figures/spectrum_all_lead_time_days_1.pdf) and we have shortened the Navier-Stokes discussion a bit to add the following discussion about the spectra in the main body of our manuscript:
> > >
> > > `Another tool to evaluate the quality of weather forecasts is to inspect the frequency pattern along a line of constant latitude. In particular, the power analysis determines the frequencies that are being conserved or lost. A model that produces blurry predictions, for example, converges to climatology (regression to the mean) and looses high-frequencies, whereas noisy model predictions with artifacts turn evident in too large power values at certain frequencies. We contrast the power spectra of the best candidate of each model class at five different lead times of one day, one week, one month, one year, and 50 years in Figure 6. The spectra at one day lead time indicate that all models start to loose power at a wavelength of 5000km and shorter, meaning that fine grained information is not well conserved in any model already after one day. GraphCast (dark blue) stands out with a comparably strong deviation from the desired frequency distribution (grey), loosing power between wavelengths of 7000 and 3000km and overshooting the verification at 2500km and below, which indicates fine grained noise patterns in the forecast, as visible in Figure 14 of the Appendix. At a lead time of seven days, the power spectrum of GraphCast greatly deviates from the verification and exceeds the plot range (see Figure 18 for the evolution of the power spectrum per model at different lead times). Also, at seven days lead time, all models start to deviate from the ground truth already at wave lengths of 11,500km, yet, in the window of 7000 and 3000km, they hardly deteriorate further, meaning they do not blur the forecasts further after the initial blurring at one day lead time, i.e., we observe no further regression to the mean, which we attribute to our 24h optimization cycle. The pattern is preserved at 31 days lead time, albeit with TFNO2D, U-Net and ConvLSTM starting to deviate stronger at very long and short wavelengths, indicating instability of these models. This instability is emphasized more at a lead time of 365 days, where FNO2D, TFNO2D, U-Net, and ConvLSTM (both on the cylinder and the HEALPix mesh) gain too much power over the entire frequency range, suggesting artifacts along all wavelengths, which proves them as physically implausible. At a lead time of 50 years, only SwinTransformer, FourCastNet, SFNO, FourCastNet, Pangu-Weather, and MeshGraphNet remain in the desired power regime. These models, excluding MeshGraphNet that imitates persistence, can therefore be considered as stable in terms of producing a physically plausible power pattern across all frequencies even at very long lead times.`
> > >
> > > We like the new content, as it complements our results adequately, underlining the stability of SFNO, FourCastNet, and Pangu-Weather, while questioning the reliability of GraphCast. In essence, we hope this analysis now better meets your expectations.

---

### Official Review · Reviewer_BnFD · 2024-11-09

**Soundness:** 2
**Presentation:** 3
**Contribution:** 2
**Rating:** 3
**Confidence:** 5

**Summary:**

The paper aims to conduct a comprehensive evaluation and comparisons of deep learning backbones for weather forecasting. Authors selected seven widely used networks and conducted a large number of experiments on both synthetic dataset and a low resolution Weatherbench dataset.

**Strengths:**

Pros:

1. Fair and comprehensive evaluations of the influence of network backbones to DLWP is important.
2. Apart from different backbones, authors also evaluate the influence of parameter numbers, which would be informative for exploring the parameter scaling law for DLWP.

**Weaknesses:**

Cons:

1. According to my experiments, tuning the parameters for weather prediction (e.g., on weatherbeach) is hard and can cause significantly different results, which hampers the reliability of the results.
2. According to my experiments, different models have different rates of convergence, which is not considered and analysized in the paper and further hampers the reliability of the results.
3. In table 1, the models saturated easily, which is not consistent with existing weather models that have more than 1B parameters. I would suggest the authors to explore model techniques to save the memory.
4. Some important works in the field are not considered and discussed, such as FengWu, FengWu-GHR, and Stormer.

**Questions:**

please refer to the weakness

---

> ### Author Response · Authors · 2024-11-28
>
> Thanks for looking into our manuscript and for acknowledging the importance of the controlled and fair experiments we present. Please find our responses to your cons in the following.
>
> **Con1 and Con2:**
> Performing exhaustive parameter searches and investigating individual convergence rates on all 179 core models is computationally not feasible, as stated in footnote 10 around lines 429–430 of our manuscript. We explored different hyperparameters when our results did not match with the literature. To test how well our default configurations trained the models, we now ran a couple of experiments with a different optimizer ([Ranger](https://github.com/lessw2020/Ranger-Deep-Learning-Optimizer)), which is reported to find suitable hyperparameter configurations for arbitrary models. Yet, our results did not change significantly. We conclude that the models in our benchmark are optimized reasonably well.
>
> **Con3:**
> Since all models converged already at 32M params, we see no reason to train 1B parameter models on the synthetic Navier-Stokes dataset. Similarly, the memory constraints do not impose limitations to our analysis, since we found the error towards which each model converges to before running out of memory.
>
> **Con4:**
> We have discussed Stormer already in our Related Work section and in our Conclusion. We initially decided not to include Stormer, ClimaX, and Fuxi as separate models in our benchmark, since they are all Transformer based, which we cover with SwinTransformer and Pangu-Weather already. We would like to point you to Reviewer F4LR, saying that `the set of models chosen for the comparison is justified well.` Thanks for pointing us to the FengWu Transformer architectures, we have added them to our Related Work.
>
> We believe our investigations provide valuable insights to the community, such as recognized by reviewer 3TqW. Kindly let us know if you are missing a statement to certain aspects of your review.

---

### Author Response · Authors · 2024-12-04
**Summary of Rebuttal**

We want to thank all reviewers and the AC for their constructive feedback and thorough consideration of our manuscript. We addressed the questions of the reviews and performed additional analyses to further improve the quality of our work. In the following, we summarize  the discussions we had with each reviewer.

## Reviewer BnFD
- **Main Concern** No individual tuning of hyperparameters for each model questions the reliability of the results.
- **Action** We have looked into the optimization of our models and found they are optimized reasonably well.
- **Comment** We have not heard back from the reviewer.

## Reviewer F4LR
- **Main Concern** Physical soundness analysis should be further extended to looking at power spectra.
- **Action** We performed power spectra analysis and included them in our manuscript, along with a thorough discussion of the results.
- **Comment** We thank the reviewer for their constructive and approachable feedback, which we could implement directly in extending analysis.

## Reviewer ijdW
- **Main Concern** Limited novelty and contributions unclear.
- **Action** We responded with a detailed list of novel contributions which the reviewer might have overseen and pointed to the sections in our manuscript, where our contributions are described.
- **Comment** We had difficulties understanding why the reviewer could not find our contributions (other reviewers did embrace our contributions and efforts) and were surprised by a repetition of the same arguments in the reviewers answer to our response. We had the impression the reviewer might not have read our manuscript and responses carefully and were hoping to receive constructive recommendations on how to improve our work.

## Reviewer 3TqW
- **Main Concern** Results on other variables beyond geopotential at 500hPa, air temperature at 2m height, and zonal wind at 10m height are missing.
- **Action** We performed additional analyses on all remaining variables, i.e., v-component of wind, temperature at 850hPa, and geopotential at 250, 700 and 1000hPa and included RMSE and ACC plots over parameters in the appendix.
- **Comment** We highly appreciate the concrete and constructive feedback of the reviewer and like the lively discussions we had, which gave us the impression that the reviewer has considered our work thoroughly.

We hope the reviewers and AC will continue in fruitful discussions and will share details about a final decision.

---

### Meta-Review · Area_Chair_pSAk · 2024-12-23

**Metareview:**

This paper is primarily experimental and compares widely used deep learning models for weather prediction.

There is one review in strong support, while the rest are critical of the work. The biggest criticisms center around lack of innovation (clarity of contribution), experimentation and in general the key learnings that are actionable.

Given the experimental nature of the work, it is important that multiple reviews recommend acceptance. Hence my recommendation is a reject.

**Additional Comments On Reviewer Discussion:**

One of the reviewers was an outlier, however it was hard for me to recommend acceptance just based on one positive review.

---

### Decision · Program_Chairs · 2025-01-22

Reject